# Dual-polarized 8-port sub 6 GHz 5G MIMO diamond-ring slot antenna for smart phone and portable wireless applications

**Yasir Fawad**[1☯], **Sadiq Ullah**[2☯], **Muhammad Irfan**[3☯], **Rizwan Ullah**[2☯], **Saifur Rahman**[3☯], **Fazal Muhammad** [4☯]*, **Abdulkarem H. M. Almawgani**[3☯], **Salim Nasar Faraj Mursal** [3☯]

**1** Telecommunication Engineering Department, University of Engineering and Technology Peshawar, Mardan, Pakistan, **2** Telecommunication Engineering Department, University of Engineering and Technology Mardan, Mardan, Pakistan, **3** Electrical Engineering Department, College of Engineering, Najran University, Najran, Saudi Arabia, **4** Electrical Engineering Department, University of Engineering and Technology Mardan, Mardan, Pakistan

☯ These authors contributed equally to this work.
* fazal.muhammad@uetmardan.edu.pk

**Data Availability Statement:** All relevant data are within the paper.

**Funding:** The authors acknowledge the support from the Deanship of Scientific Research, Najran

## Abstract

This manuscript presents high performance dual polarized eight-element multiple input multiple output (MIMO) fifth generation (5G) smartphone antenna. The design consists of four dual-polarized microstrip diamond-ring slot antennas, positioned at corners of printed circuit board (PCB). Cheap Fr-4 dielectric with permittivity 4.3 and thickness of 1.6mm is used as substrate with overall dimension of $150 \times 75 \times 1.6\ mm^3$. In mobile system due to limited space mutual coupling between nearby antenna elements is an issue that distort MIMO antenna performance. Defected ground structure is used to control coupling. The defected ground structure has advantages like ease of fabrication, compact size and high efficiency as compare to other techniques. Less than 30dB coupling is achieved for adjacent elements. The -10 dB impedance bandwidth of 700 MHz is achieved for all radiating elements ranging from 3.3 GHz to 4.1 GHz. The value is about 900MHz for -6dB. The proposed antenna offers good results in terms of fundamental antenna parameters like reflection coefficient, transmission coefficient, maximum gain, total efficiency. The antenna achieved average gain more than 3.8dBi and average radiation efficiency more than 80% for single dual polarized element. The antenna provides sufficient radiation coverage in all sides. The MIMO antenna characteristics like diversity gain (DG), envelope correlation coefficient (ECC), total active reflection coefficient (TARC) and channel capacity are calculated and found according to standards. Furthermore, effect of user on antenna performance in data-mode and talk-mode are studied. Proposed design is fabricated and tested in real time. The measured results shows that proposed design can be used in future smartphones applications. The design is compared with some of the existing work and found to be the best one in many parameters and can be used for commercial use.

University. Kingdom of Saudi Arabia, for funding this work under the Research Groups funding program grant code number (NU/RG/SERC/12/9). The funders had no role in study design, data collection and analysis, decision to publish, or preparation of the manuscript.

**Competing interests:** The authors have declared that no competing interests exist.

## Introduction

In modern day, the interest in MIMO wireless communication technology is increased due to key features of MIMO technology like highly improve wireless link reliability, transmission capacity and data rates of wireless system through multi-path transmission and reception [1]. For 5G research societies now concentrating on achieving high data rates at low cost. It is predicated that the 5G data rate will be thousand times higher than the current communication technologies [2]. The number of users and connected devices are increasing rapidly. 5G wireless network employed MIMO technique to support low latency, high data rates and massive user connectivity [3]. MIMO technology is the best solution for achieving high data rates [4]. Standard MIMO antenna consist of two to four radiating elements in single suite while massive MIMO consist of large number of antennas. $2 \times 2$ MIMO antenna is deployed for current 5G applications, while for future 5G massive MIMO antenna with greater number of elements is expected, as large number will make the system more resilient to signal interference and intentional jamming [5]. Printed antenna is most suitable antenna for 5G among other antenna because of its characteristics like ease of fabrication and low cost and capability to easily installed on small devices like mobile phones and portable devices [6]. Some of the techniques used for mutual coupling reduction are defected ground structure [7]., dielectric resonator antenna [8], complementary split ring resonator (CSSR) [9], decoupling network [10], neutralization lines [11], parasitic or slot elements [12], electromagnetic bandgap structures (EBG) [13, 14] and metamaterials [15–17]. EBG structure and metamaterials are complex and require large area. Resonators have complex fabrication procedure [18]. Some techniques lead to narrow bandwidth and low antenna gain and efficiency. However, Defected ground structure is taken due to its ease of fabrication and low circuit complexity. One unique characteristics of DGS is slow wave propagation which leads to miniaturization of antenna [19]. In MIMO technology the problem of mutual coupling greatly influences the performance of antenna, mutual coupling arises by placing multiple elements in limited space in MIMO system [20]. A comparative study between different mutual coupling reduction techniques is done by the author of [21] in detail. According [22] slot in ground plan is a good technique for reducing mutual coupling. According to the author of [23] compactness, ease of fabrication, wide band and low mutual coupling are needs of cellular communication. To achieve the aforementioned characteristics and desired data rates at least six to eight element MIMO antenna is required [23]. So, MIMO antenna is urgent demand of 5G communication technology and smart phones applications. Recently several antenna designs [23–31] have been presented for 5G sub 6GHz band mobile terminals addressing these issues. In [24] an orthogonally dual polarize antenna is proposed but the antenna gain is only 3dB and antenna also has high mutual coupling. In [25] wide impedance bandwidth is achieved on the expense of efficiency and isolation. The work presents in [26] has low efficiency, narrow bandwidth with 5.3 substrate thickness. In [27–29] antenna presented have high mutual coupling and low gain. The design in [30, 31] has narrow bandwidth and low isolation between antenna elements. The minimum number of antenna MIMO elements that works in sub 6 GHz band is six to eight [32]. This is novel shape eight-port mobile phone antenna design that features small dual-polarized radiation elements and offer large impedance bandwidths for 5G applications. Several countries have defined their own standard for 5G communication. World radio communication conference (WRC) allocate C-band for 5G communication ranging from 3.4 to 3.6 GHz in 2015, while China and European Union select C-band and 3.4 to 3.8 GHz band respectively [33]. The UK's Ofcom has recommended 3.6 GHz as potential frequency band for sub-6 GHz 5G cellular networks [34], and this antenna is made to function at that frequency. In [35] a

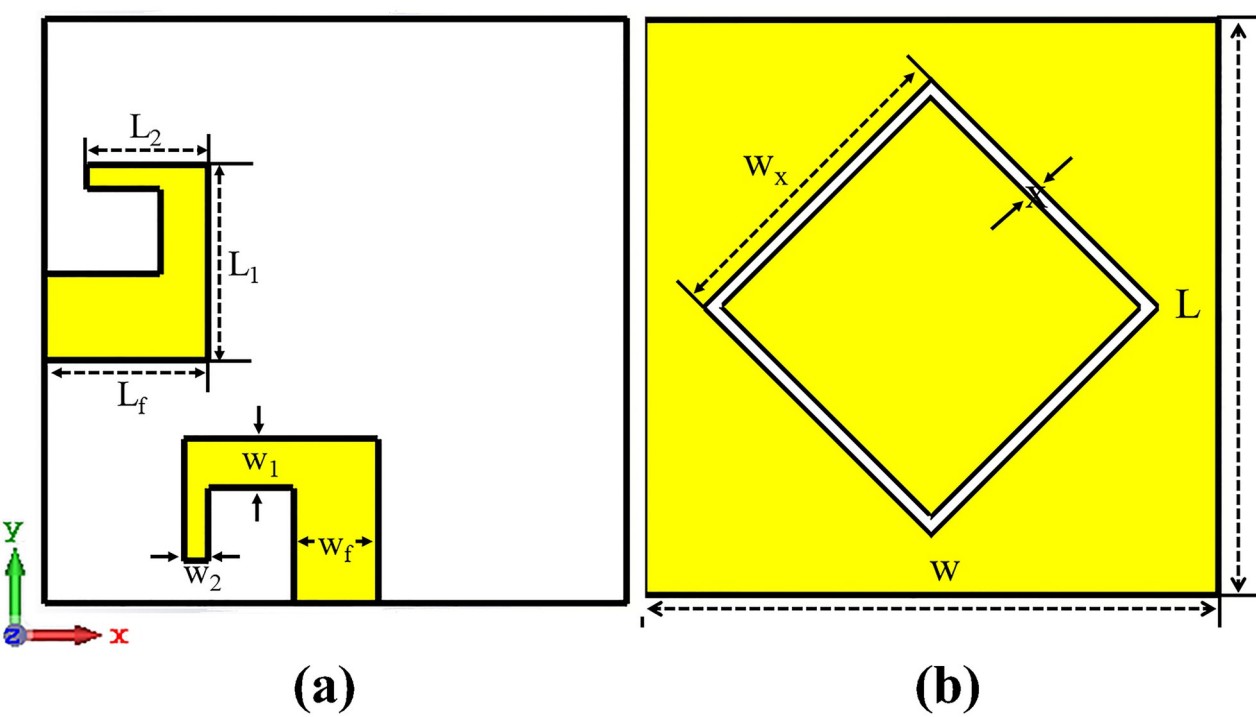

**Fig 1. Dual polarized single element.** A: Front View. B: Back View.

flexible sub 6 GHz 5G MIMO antenna has been proposed achieving 92% impedance bandwidth on the expense of gain and efficiency. The author of [36] also proposed MIMO antennas for mobile terminal achieving excellent results but none of them has operating band in sub 6GHz band both are mm wave antennas, which are not suitable for current 5G mobile phones. The antennas mentioned above have some limitation like non-planar configuration, complex structure, high cost, bulky and large size and some are simply not suitable for current 5G smartphones. This article focused on a simple, low cost and low weight antenna that is operating in desired current 5G operating band. The printed circuit board's corners are home to four double-fed, dual-polarized antenna elements in design arrangement. On various sides of mobile phone PCB, antenna elements have high isolation and wide bandwidth, which provide polarization diversity characteristics. As a result, the design is capable of supporting various polarizations in addition to offering complete radiation coverage. The investigation of antenna properties using computer simulation technology (CST) software.

**Table 1. Optimized Values of Antenna Design Parameter.**

| Parameter | Value (mm) | Parameter | Value (mm) |
|---|---|---|---|
| Width of Substrate ($W$) | 24 | Width of Arm 1 ($W_1$) | 1 |
| Length of PCB ($L_s$) | 150 | Length of Arm 2 ($L_2$) | 5 |
| Width of PCB ($W_s$) | 75 | Width of Arm 2 ($W_2$) | 2 |
| Height of Substrate($H$) | 1.6 | Width of Slot 1 ($W_x$) | 10 |
| Length of Feed ($L_f$) | 7 | Gap ($X$) | 0.5 |
| Width of Feed ($W_f$) | 3 | Length of Arm 1 ($L_1$) | 3.5 |

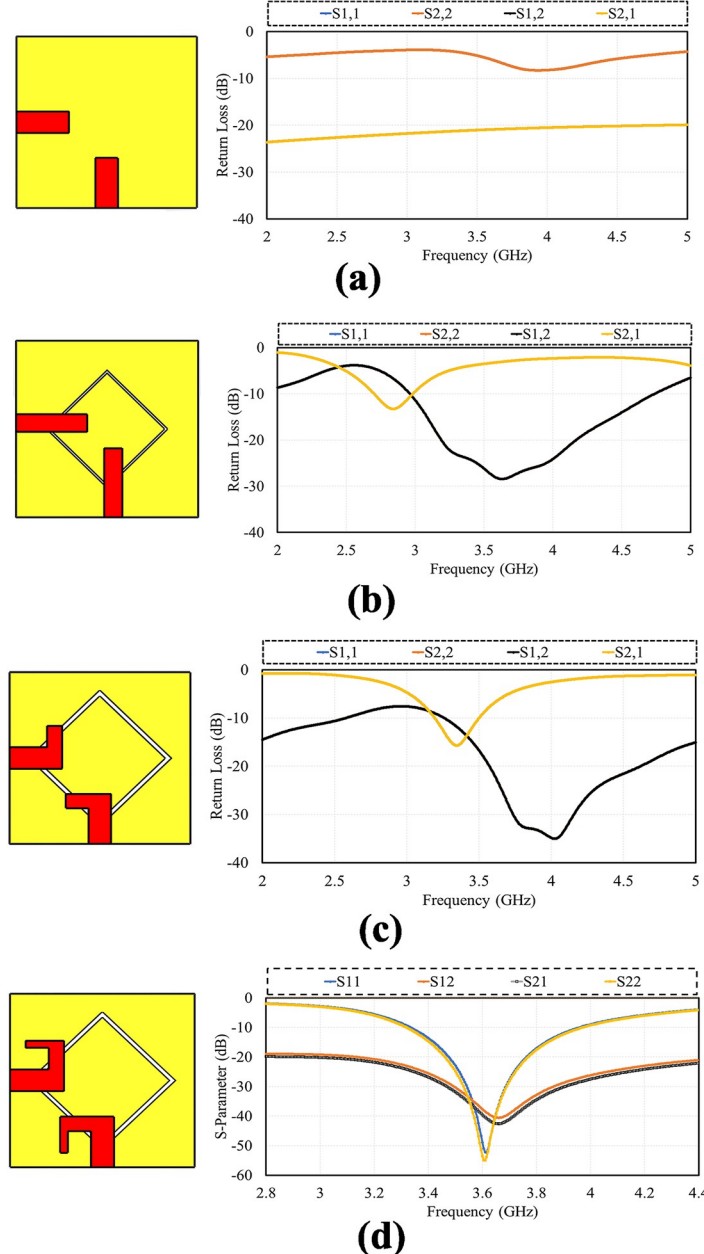

**Fig 2. Design steps.** A: Feed with Full ground. B: Feed with Diamond Slot. C: L-shape Feed with diamon slot. D: Proposed Design.

The single element and MIMO mobile phone antenna are examined in terms of fundamental antenna parameters like S-parameters, Gain, total efficiency, radiation pattern, ECC, TARC, Diversity Gain and channel capacity basic characteristics of the single-element and its MIMO design are examined. The proposed design has been constructed, measured, and the comparison of the results with those from simulations reveals good agreement. Investigations are also conducted into how user behavior affects the performance of the antenna in data and talk mode.

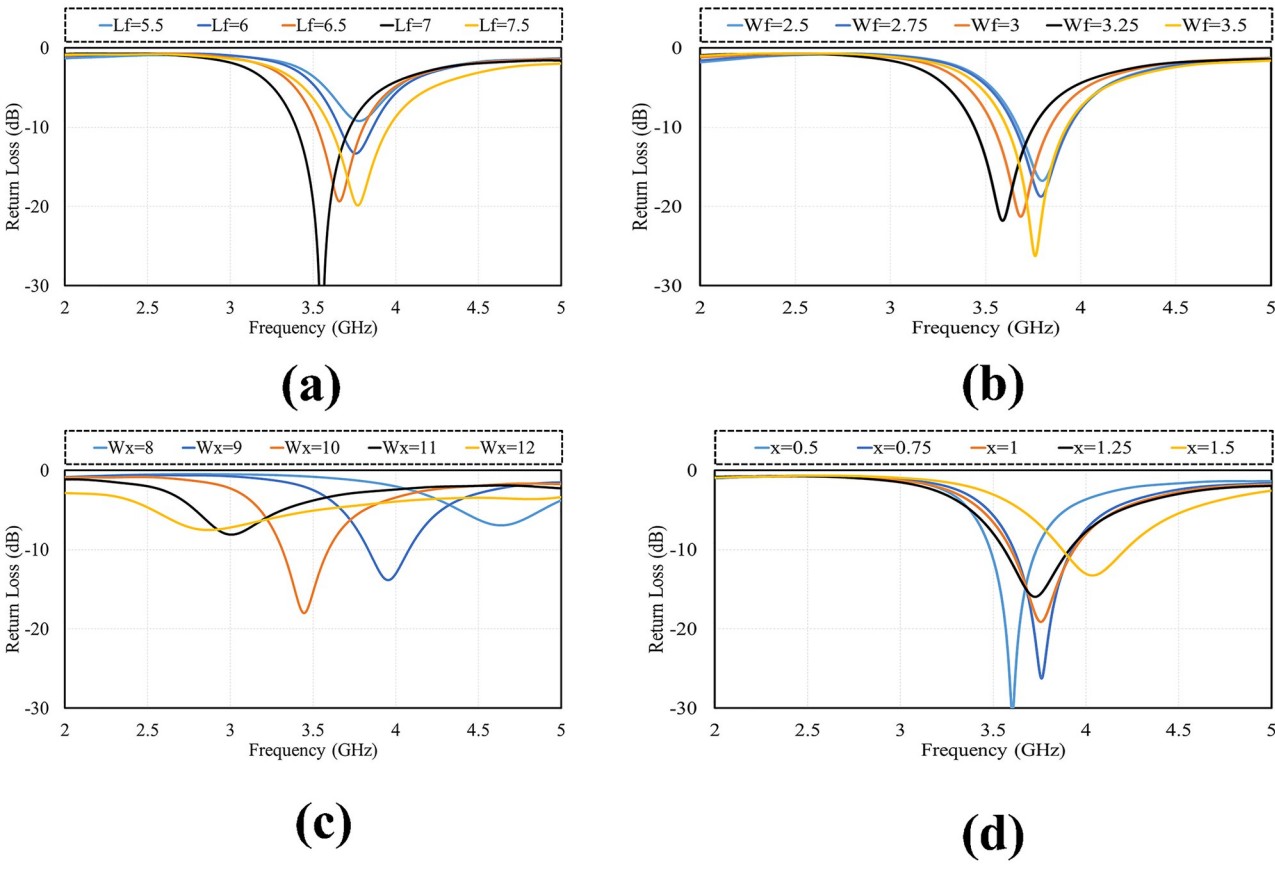

**Fig 3. Parametric analysis.** A: $L_f$. B:$W_f$. C:$W_x$. D:X.

## Anetnna design methodology and results

### Dual polarized single element antenna

Here, the design of single element dual-polarized antenna and result analysis of simulated and measured results is discussed. In figure below the configuration of dual-polarized antenna is presented. Fig 1 shows the front and back view of the proposed antenna design. The antenna radiating elements are mounted on relatively cheap Fr-4 dielectric substrate with 4.3 relative permittivity and 0.025 loss tangent having height of 1.6 mm. Two microstrip feed lines as a radiating ground plane with diamond-ring slot are included in antenna configuration. The dimension of single element dual polarized is $24 \times 24 \times 1.6 \ mm^3$. The optimized value of the proposed MIMO design is listed in below Table 1.

The aim of the research is the design of a small MIMO antenna that can be easily integrated into a smartphone's PCB with wide band and having dual polarization capabilities and low mutual coupling. This is accomplished by using microstrip patch antenna with slot in ground plane which increase isolation and wide the antenna bandwidth. Slot antenna is used because of its attractive features like compactness, light wight and ease of fabrication and integration with radio frequency circuits. The width of slot determines operating frequency. The circumference of ring needs to match the dielectric wavelength at the operating

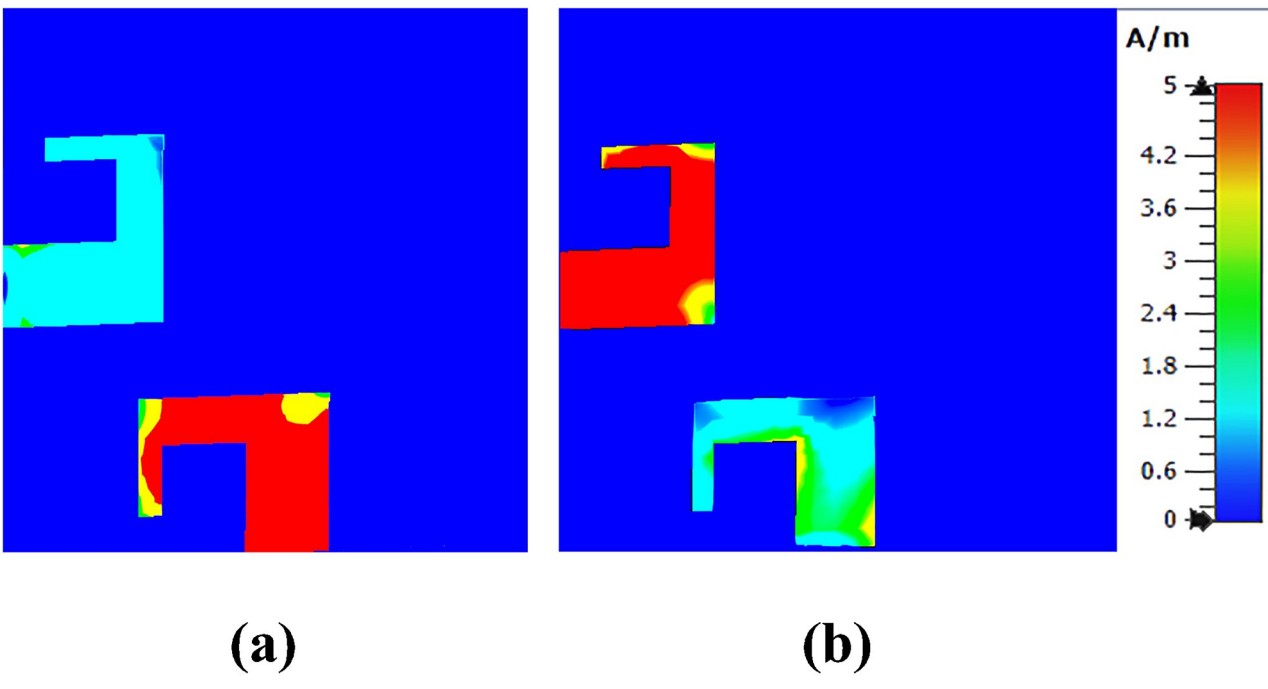

**Fig 4. Surface current distributions.** A: Port1. B: Port2.

frequency as Eq (1).

$$\frac{wx}{2} + g = \lambda \tag{1}$$

The dual polarization of antenna is achieved by placing radiating elements orthogonal to each other so that they orthogonally polarized. Dual polarization also increase isolation between antenna elements as both elements fed differently. Microstrip feed lines are used for feeding the orthogonal antenna elements. Microstrip feeding techniques offers features like

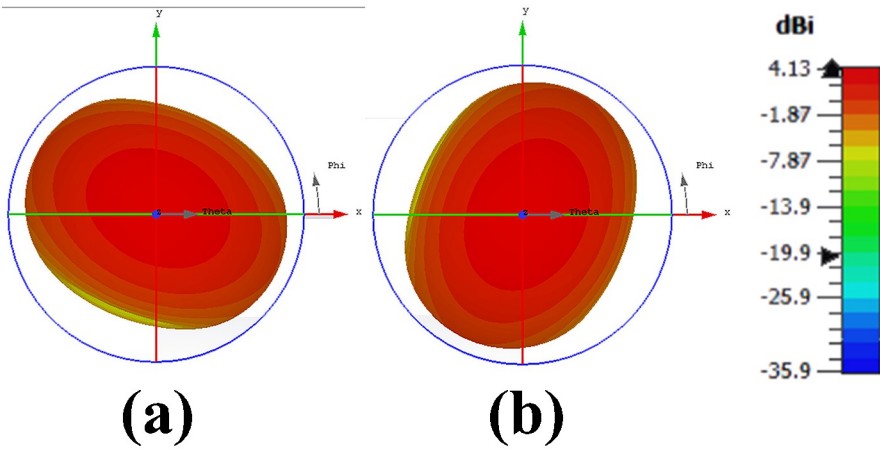

**Fig 5. 3D radiation pattern.** A: Port1. B: Port2.

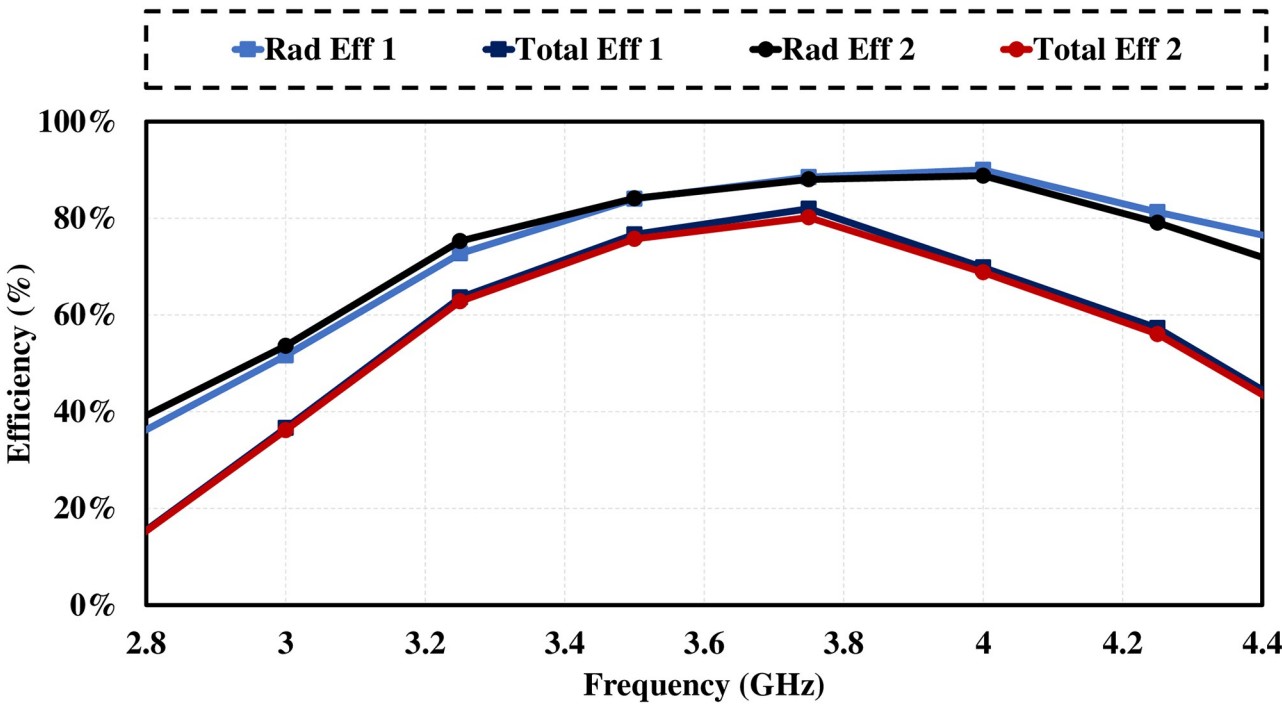

**Fig 6. Gain and efficiency.** Efficiency.

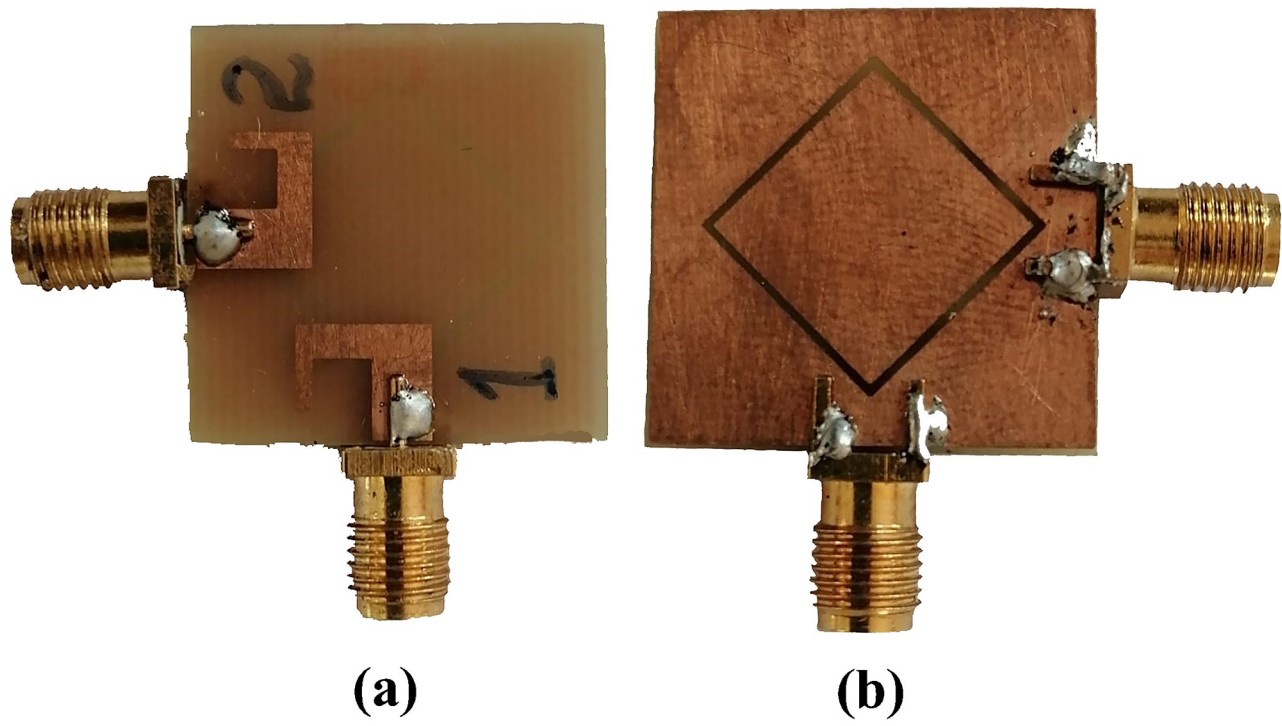

**Fig 7. Fabricated prototype.** A: Front View. B: Back View.

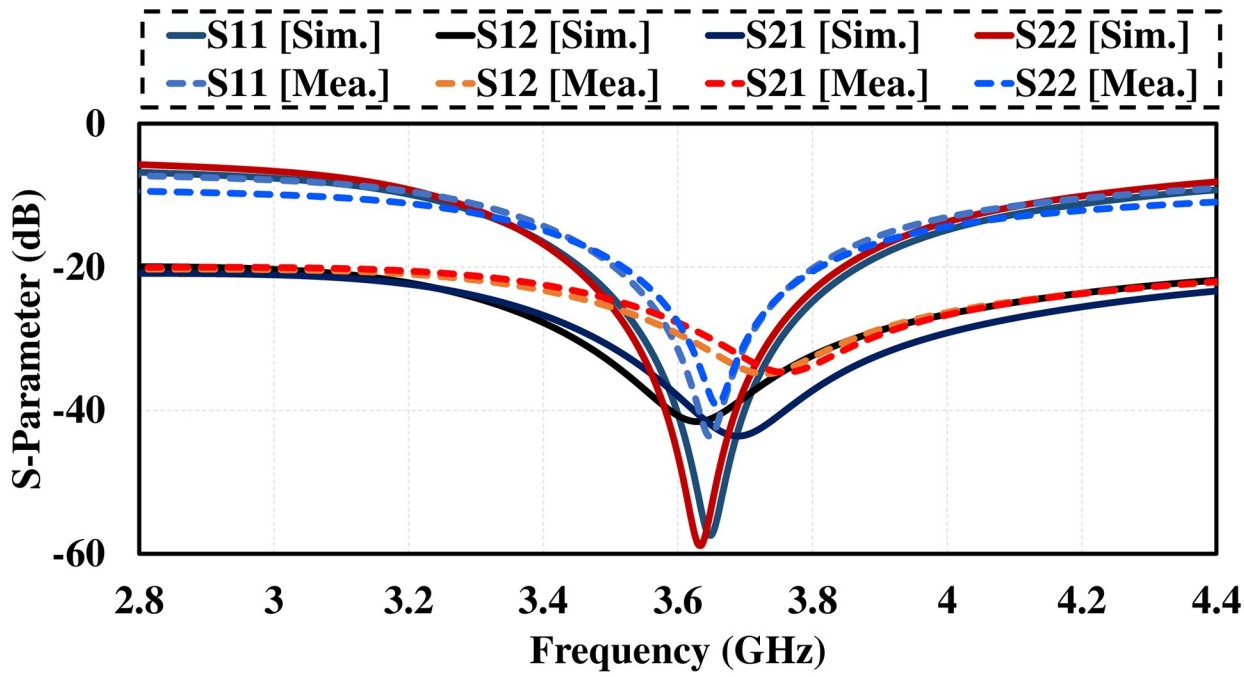

**Fig 8. Measured and simulated $S_{11}$ of dual polarized antenna.**

simplicity, miniaturization, low cost, planar compatibility, high integration which is focus of this research. The characteristic impedance (Zo) of microstrip feed line can be calculated as Eq (2)

$$Z_0 = \frac{87}{\sqrt{\varepsilon_r + 1.41}} \cdot \frac{W}{H} \tag{2}$$

Where effective height of the feed line can be calculated as Eq (3)

$$H_{\text{eff}} = H \cdot \varepsilon_{\text{eff}} \tag{3}$$

$\varepsilon_{\text{eff}}$ is given by Eq (4)

$$\varepsilon_{\text{eff}} = \frac{\varepsilon_r + 1}{2} + \frac{\varepsilon_r - 1}{2} \cdot \left(1 + 12 \cdot \left(\frac{H}{W}\right)\right)^{-0.5.} \tag{4}$$

The configuration of various structures studied in the designing process of proposed dual polarized antenna alongside with their S-parameter are displayed in Fig 2. In first step a straight microstrip-feed with full ground is taken, then diamond-ring shape slot is etched in the ground plane for increasing the impedance bandwidth and reduce coupling, the size of radiating element increases to shift the frequency to L-shaped radiating element with diamond slot and at the last stage the proposed design is made which has wide bandwidth, high isolation and low mutual coupling operating at 3.6 GHz. The four designing stages are presented in Fig 2A to 2D respectively.

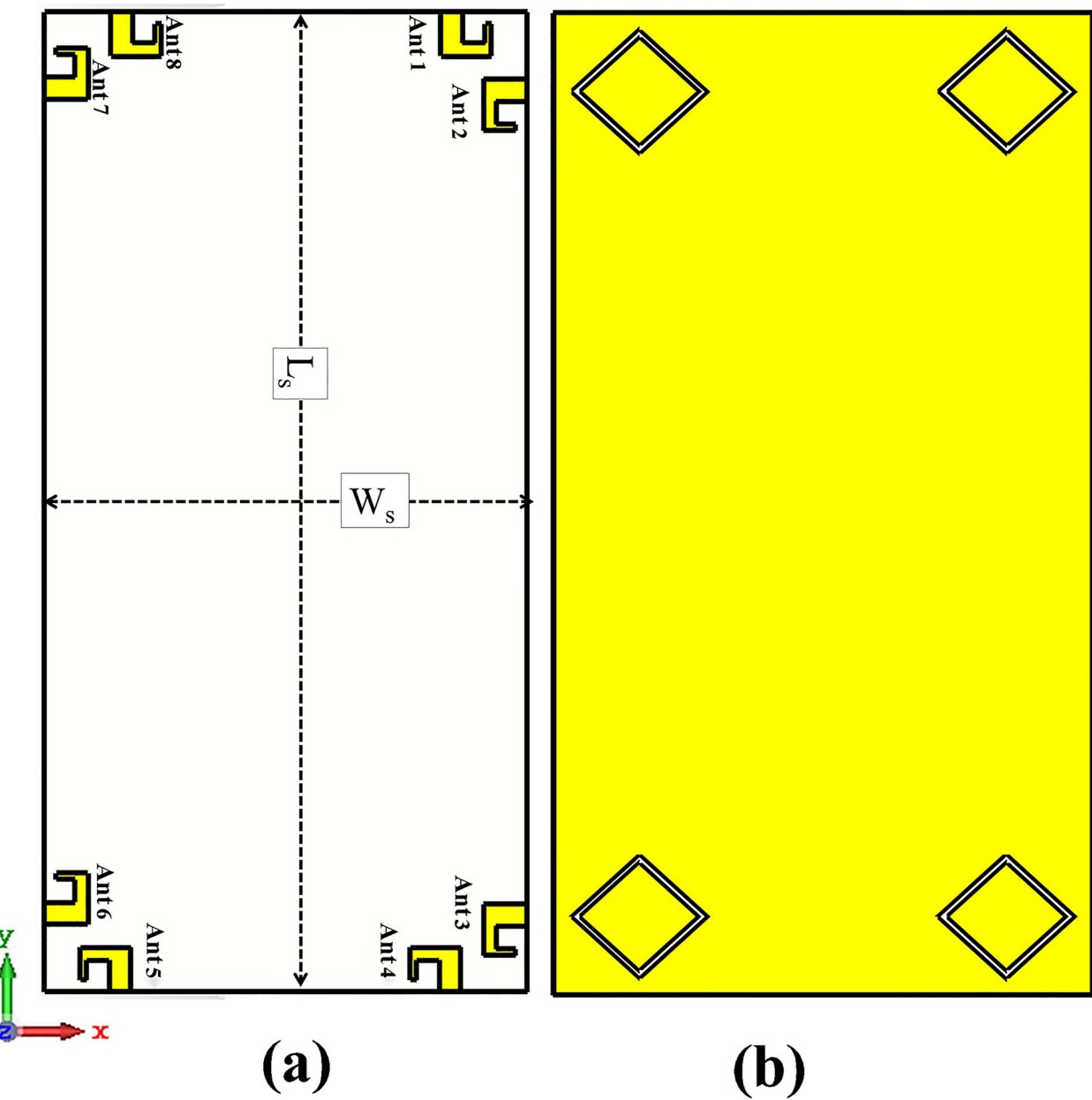

**Fig 9. Proposed smartphone antenna.** A: Front View. B: Back View.

Reflection coefficient (S11) of various antenna design parameters like Lf, Wf, Wx and x are demonstrated in Fig 3. The effect of changing the feed length (Lf) on the resonance frequency presented in Fig 3A, when its size 5.5 to 7.5mm, the antenna impendence matching varies with no shift in resonant frequency. The effect of feed width is demonstrated in Fig 3B. The resonant frequency shifts from high to low as the width of feed (Wf) increase from 2.5 to 4.5. Fig 3C demonstrates the variation in resonant frequency as the radius of diamond slot (Wx) changes. Central frequency varies from high to low as the radius changes from 8 to 12. Fig 3D illustrates the effect of slot width on resonant frequency with increase in size the frequency shifts form low to high.

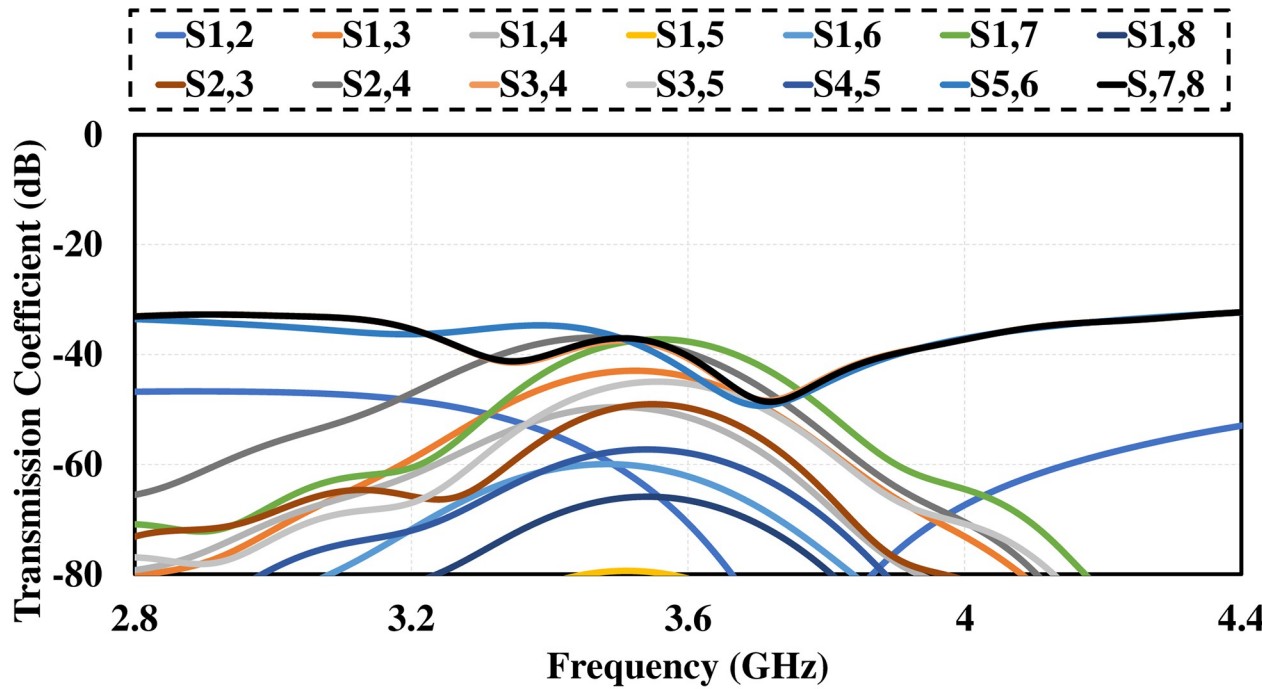

**Fig 10. Simulated S-parameter.** A: Reflection Coefficient. B: Transmission Coefficient.

Fig 4 demonstrates antenna's surface current distributions at 3.6 GHz. Surface current of an antenna shows magnitude and pattern of current that is flowing on the surface of antenna element. Radiation pattern is determined from distribution of current [33]. Well distributed current is cause of good radiation pattern. Figure clarifies the currents are mostly concentrated near the ring slot in ground plane and both antennas are independent of each other which means high isolation between antenna elements. Fig 5 shows antenna's 3D radiation patterns of both antenna elements (Port1 & Port2). As demonstrated, antenna exhibits orthogonally polarized radiation patterns with more than 3.5 dB of realized gain at whole operating band. Dual-polarized antenna's maximum gain plot against frequency and radiation characteristics in terms of radiation efficiency and overall efficiency are shown in Fig 6. The antenna has significant total efficiency, as can be observed. In whole operation band, 70-80% of the radiation efficiency is obtained. It is evident that antenna displays nearly identical radiation and overall efficiency.

The photograph of fabricated prototype of proposed antenna design (single element) and the measured reflection and transmission coefficient are given in Fig 7. the measured results are almost identical with simulated results with very little variations due to experimental errors as illustrated in Fig 8.

### Eight element MIMO smart phone antenna

Smart phone 8 by 8 antenna design on 75mm by 150 mm PCB is illustrated in Fig 9. Four dual polarized antenna elements are positioned at the four corners of PCB. The antennas are far enough from each other to reduce mutual coupling and increase isolation. The layout also shows that there is enough space to insert additional mm wave antennas in future on same

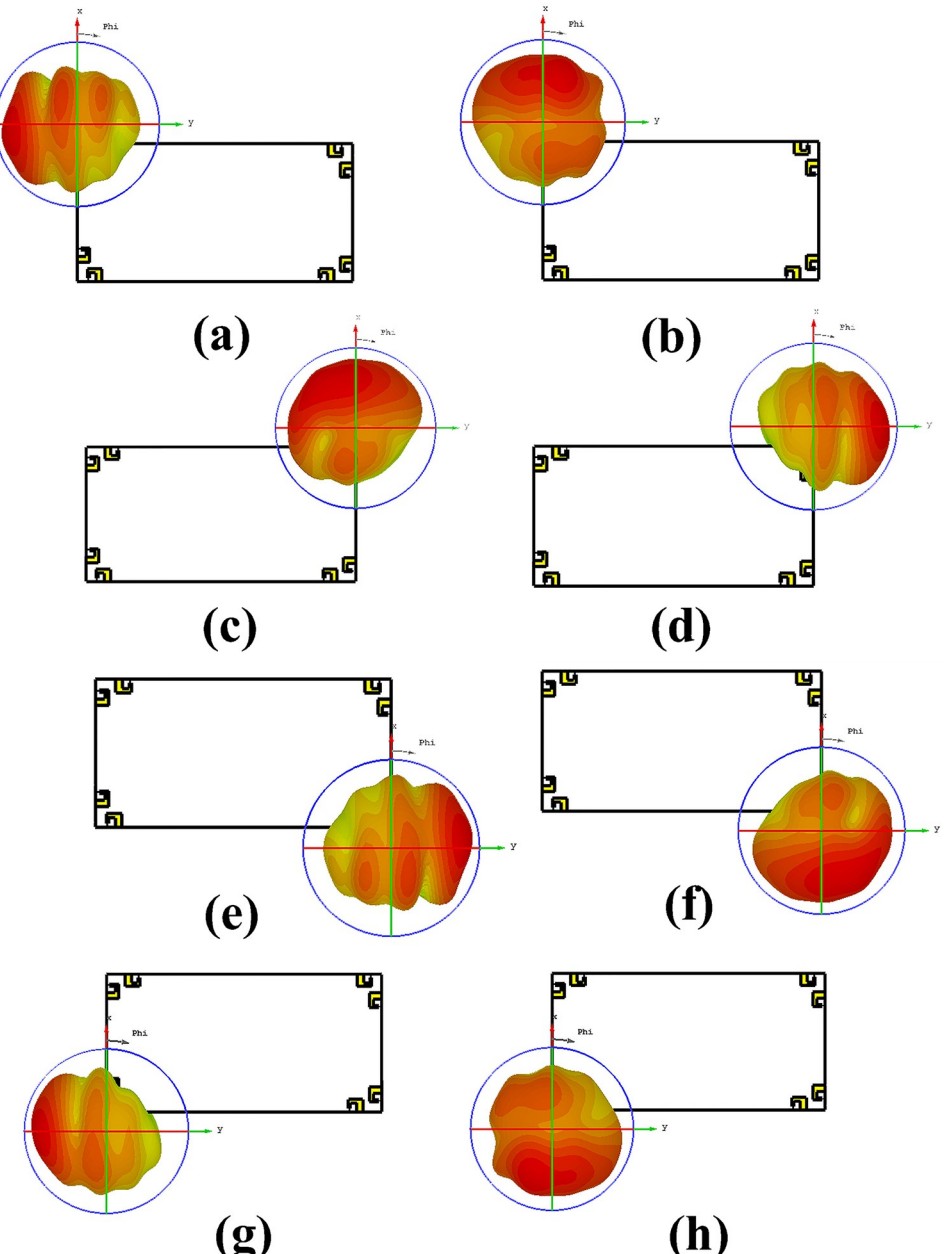

**Fig 11. 3D radiation pattern.** A: Antenna 1. B: Antenna 2. C: Antenna 3. D: Antenna 4. E: Antenna 5. F: Antenna 6. G: Antenna 7. H: Antenna 8.

PCB covering mm-wave 5G bands. Fig 10 shows the simulated S-parameter of the design over its operating band, including reflection coefficient ($S_n n$) and transmission coefficient ($S_n m$). The figure shows that the suggested antenna design for smart mobile phone exhibits good return loss, wide bandwidth, high isolation, low mutual coupling and polarization diversity. Each of the four dual polarized radiating elements offers comparable performance.

Fig 11 demonstrates the radiation pattern of proposed 8 by 8 MIMO antenna. The radiation pattern of all the elements in different direction covering all area around, means that

(A)

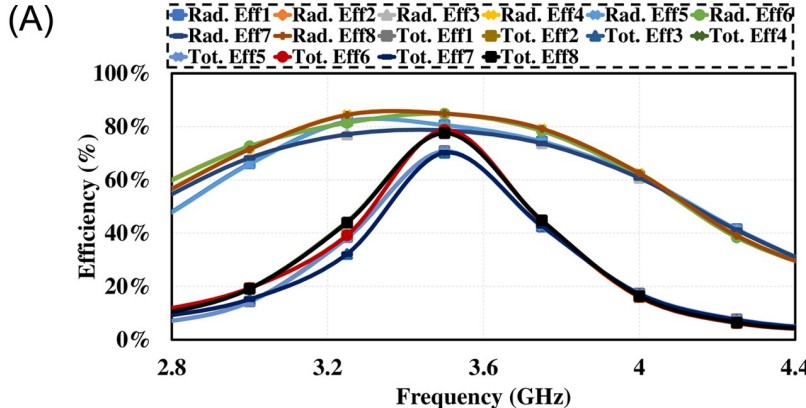

(B)

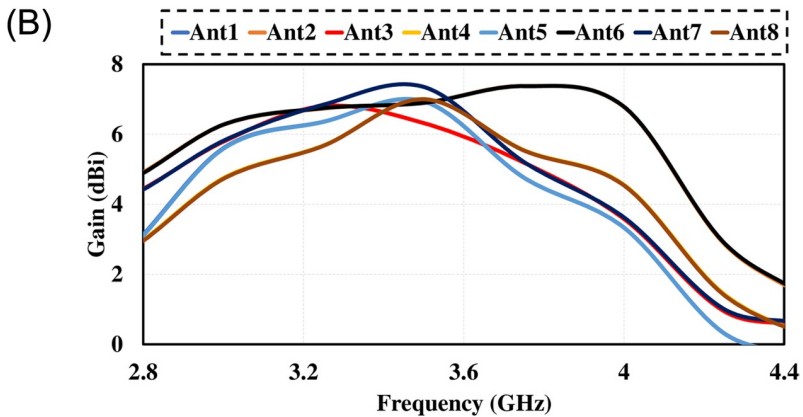

**Fig 12. Simulated results.** A: Maxmimum Gain. B:Efficiency.

antenna offers good radiation coverage and are best for multipath transmission and reception with low interference making it best for mobile phones. Results demonstrates that antenna has good radiation properties, high gain and total efficiencies in the desired frequency band. The simulated gain against frequency plot and the total and radiated efficiencies are given in Fig 12.

The photograph of fabricated prototype of 8 by 8 smart phone antenna is illustrated in Fig 13. the prototype is fabricated on lossy and cheap Fr-4 dielectric of 1.6 thickness. The PCB used to build the cell phone antenna has an overall dimension of 75 × 150 × 1.6 mm3. The fabricated design is tested and the measured S-parameter results are presented in Fig 14. the measured results shows that antenna is best for mobile phones in real world scenarios as it offers wide impedance bandwidth and high isolation with sufficient total efficiency and gain of more than 7dB in the entire band. There are some variations in measured results, errors in fabrication, feeding and experimentation procedure are the reasons for these certain inconsistencies in measurement.

The measured and simulated 2D polar plots (E & H-plane) of radiation pattern are compared and illustrated in Fig 15.

ECC, DG, TARC and channel capacity (CC) are some crucial factors that need be taken into account in MIMO antennas to find that MIMO system function properly. ECC between

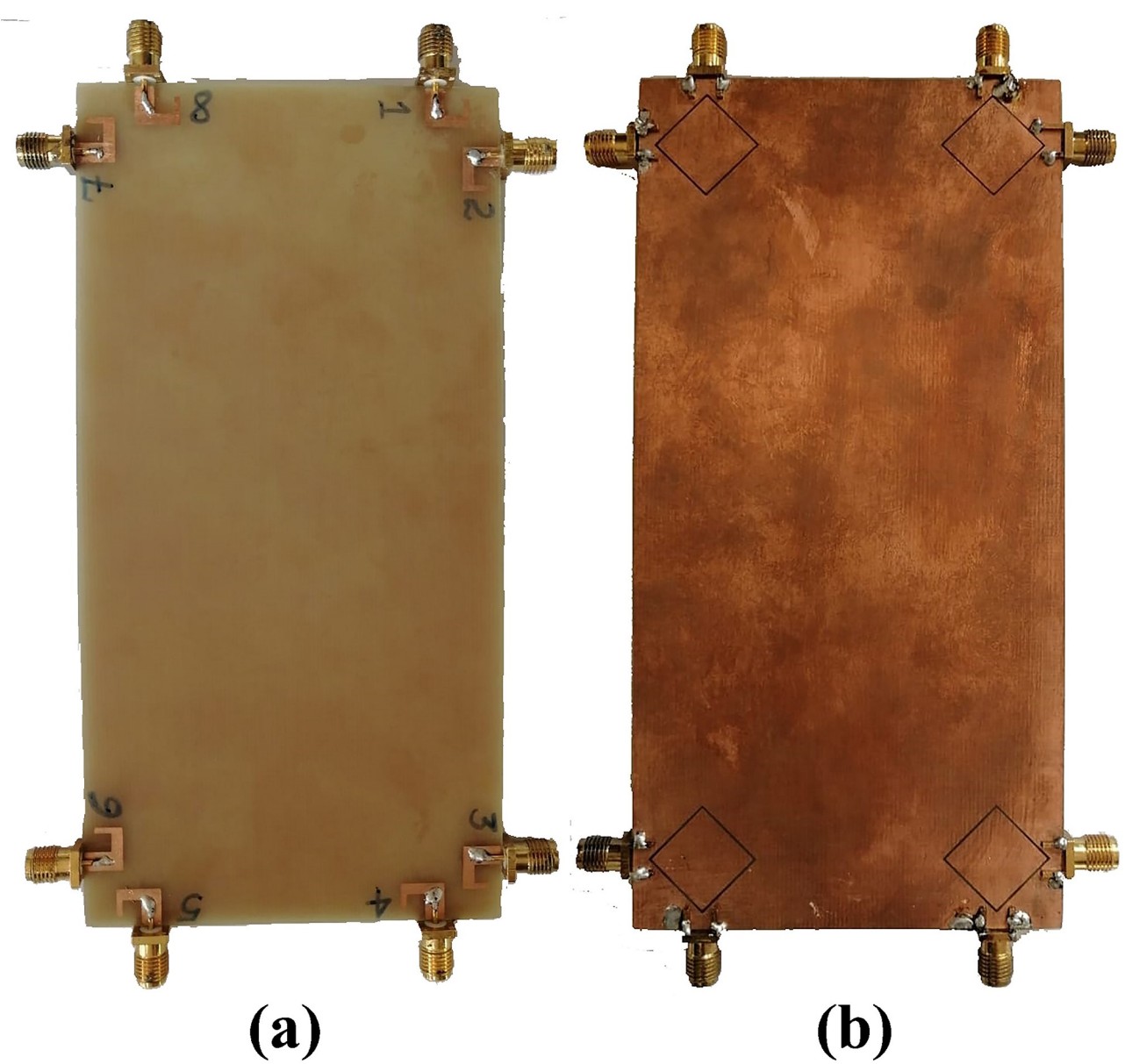

**Fig 13. Fabricated prototype.** A: Front View. B: Back View.

$i^{th}$ and $j^{th}$ element of m by n MIMO antenna can be determined from antennas S-Matrix using equations Eq (5). Diversity Gain in terms of ECC can be calculated from Eq (6).

$$|p_{ij}|^2 = \frac{||S_{ii}^* S_{ij} + S_{ji}^* S_{jj}||}{\sqrt{(1 - |S_{ii}|^2 - |S_{ji}|^2)(1 - |S_{jj}|^2 - |S_{ij}|^2)}}, \tag{5}$$

$$DG = 10\sqrt{1 - \text{ECC}^2} \tag{6}$$

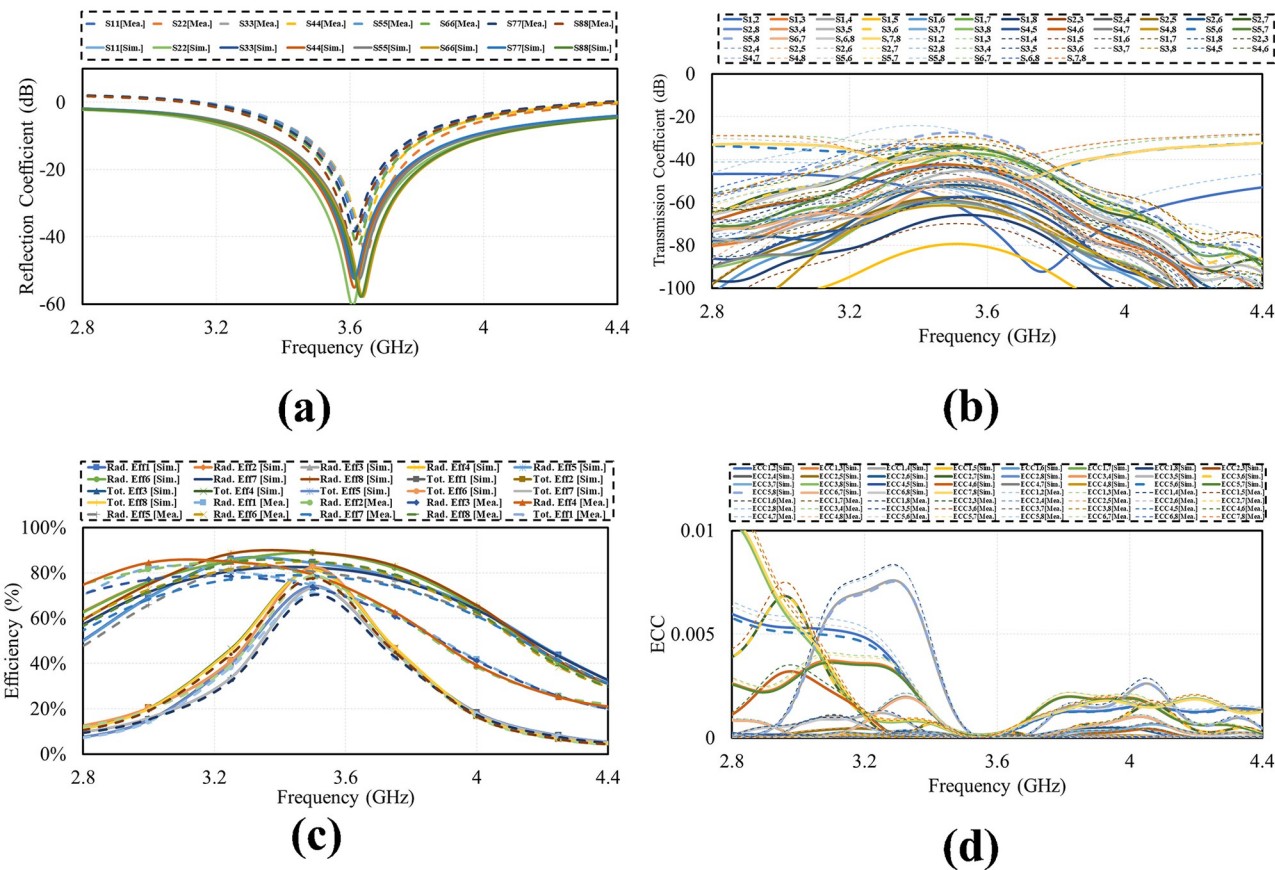

**Fig 14. Measured results.** A: Reflection Coefficient. B:Transmission Coefficient.

ECC of antenna described about how much radiation pattern of two elements of a MIMO system are independent of each other. If on element is vertically polarized and a near-by second element is horizontally polarized then these elements would have a zero correlation between them. Correspondingly, if one antenna only radiated energy towards the ground and the other antenna element has a radiation pattern towards the sky totally opposite the ECC between these antennas would also be zero. Polarization of the antenna, shape of radiation pattern and relative phase of fields between two elements of a MIMO antenna or taken into consideration when calculating ECC of MIMO system. Envelope correlation coefficient of MIMO system can be calculated using S-matrix via Eq 1. Diversity Gain can define as amount of transmission power that can be reduced by introducing technique into MIMO antennas for diversity without degrading antennas performance. DG is mostly expressed in decibels (dB) and sometime as power ratio. Soft handoff gain is an example of diversity gain. The following Fig 16 shows the ECC and DG of MIMO system for 5G communication.

An important parameter for assessing performance of MIMO antenna is TARC. It can be calculated from Eq (7). Figs 17 and 18 show TARC and channel capacity of proposed MIMO system. TARC value of the proposed antenna is less then -30dB at 3.6GHz. from the obtained results it is concluded that the proposed design is suitable for MIMO applications. channel capacity is another important parameter which shows the maximum achievable data rates by

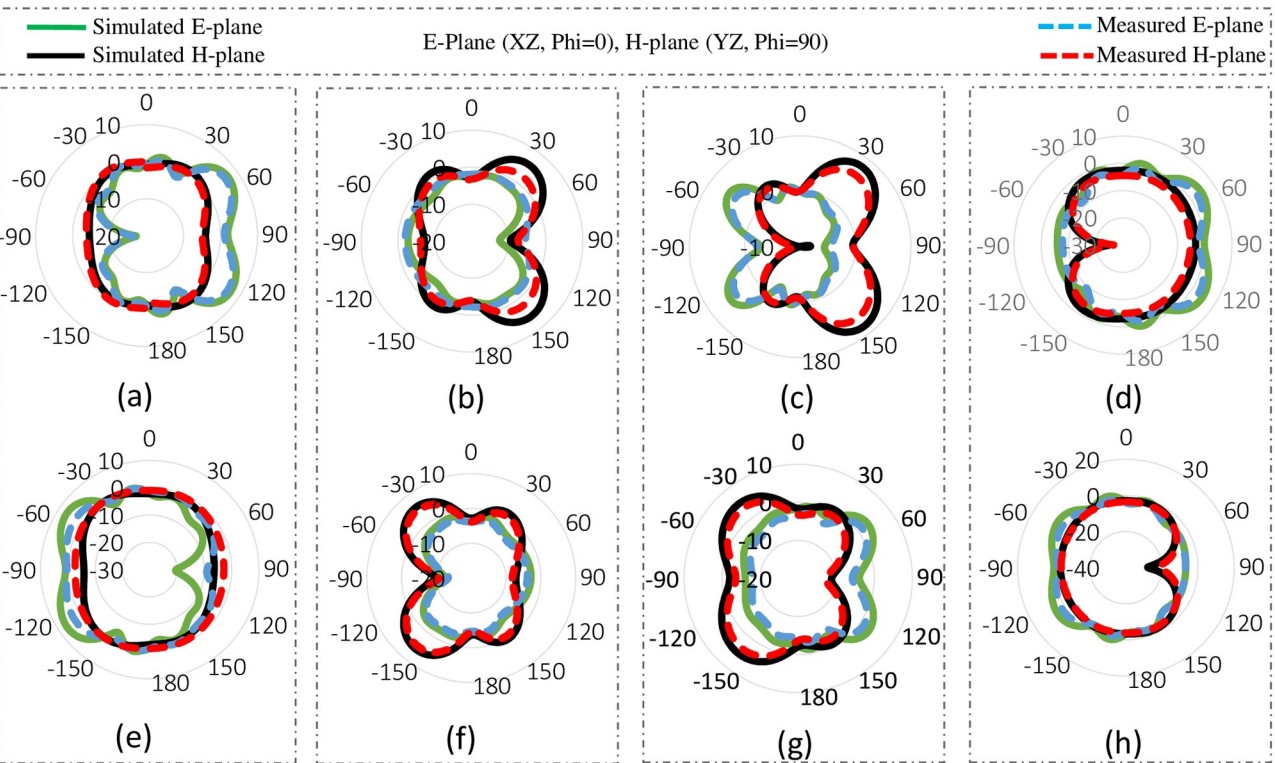

**Fig 15. Polar plots.** A: Antenna 1. B: Antenna 2. C: Antenna 3. D: Antenna 4. E: Antenna 5. F: Antenna 6. G: Antenna 7. H: Antenna 8.

MIMO system. channel capacity loss indicates the reduction in channel capacity of MIMO system as compare to ideal data rates. the ideal data rates for 8 by 8 MIMO system is 44 bps/Hz [36]. channel capacity of MIMO antenna can be calculated using Eq (8). The proposed design achieve more than 43 bps/Hz can be observe in Fig 19

$$\text{TARC} = \sqrt{\frac{(S_{mm} + S_{mn})^2 + (S_{nm} + S_{nn})^2}{2}} \qquad (7)$$

$$C = \log_2(\det(\mathbf{I} + \text{SNR} \cdot \mathbf{HH}^{\mathsf{H}})) \qquad (8)$$

The detailed comparison of proposed work with already present work in literature is given in Table 2. The proposed work shows good results specially in isolation reduction. The proposed antenna shows greater mutual coupling reduction properties and efficiency at the expense of bandwidth for the same standard mobile phone dimensions. The ECC of the antenna is very low which shows antenna's MIMO behavior. The Table 2 shows the comparison of some existing prototypes with the proposed work in some important parameters like size, bandwidth, efficiency, isolation and ECC.

## User's body impact on antenna performance

Effect of human (users) body tissues on proposed design performance is studied considering three common scenarios i.e., Data mode in which two modes single hand mode (SHM) and dual hand mode (DHM) and talking mode (TM). The placement of mobile phone MIMO

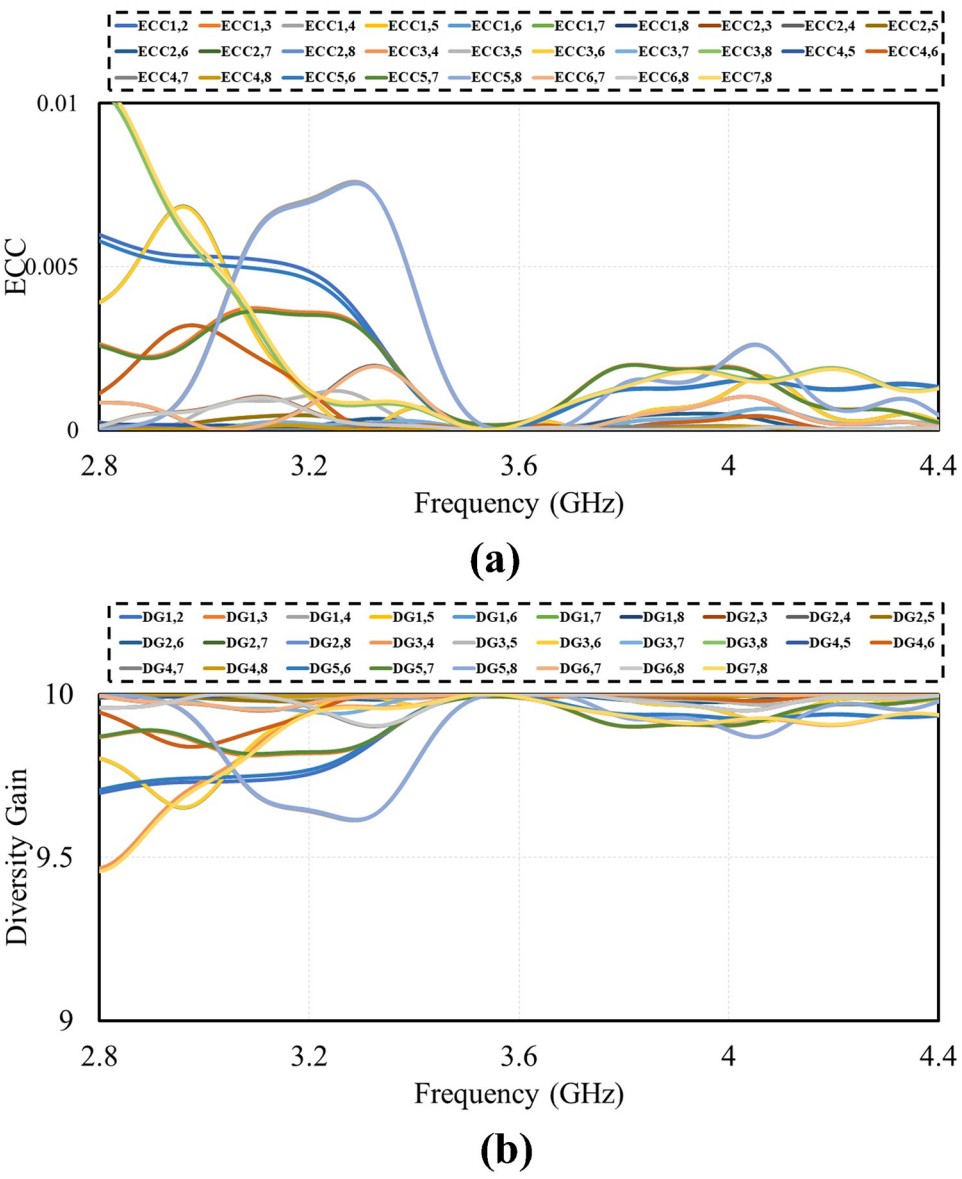

**Fig 16. MIMO results.** A: ECC. B: DG.

antenna in different scenarios is illustrated in Fig 19. The simulated reflection, transmission coefficient, efficiency and ECC in the vicinity of user's body. The simulated results (reflection coefficient, tranmission coefficient, radiation efficiency and ECC) of all the three scenarios SHM, DHM and TM are presented in the Figs 20–22 respectively. It is observed from the figures that the user body tissues has impact on antenna performance. The reflection coefficient of the antennas that are directly touched by the body are slightly shift towards higher frequency. The isolation between elements increase due to the absorption of energy by users head and hands. The Efficiency of the different modes are about 50% to 60%. The ECC has not effected much in all the three scenarios. The reflection coefficient shows that antenna can be used for 5G sub 6 GHz applications, as central frequency is in required band of 3.6 GHz. A critical issue called specific absorption rate (SAR) also needs to be identified for antenna

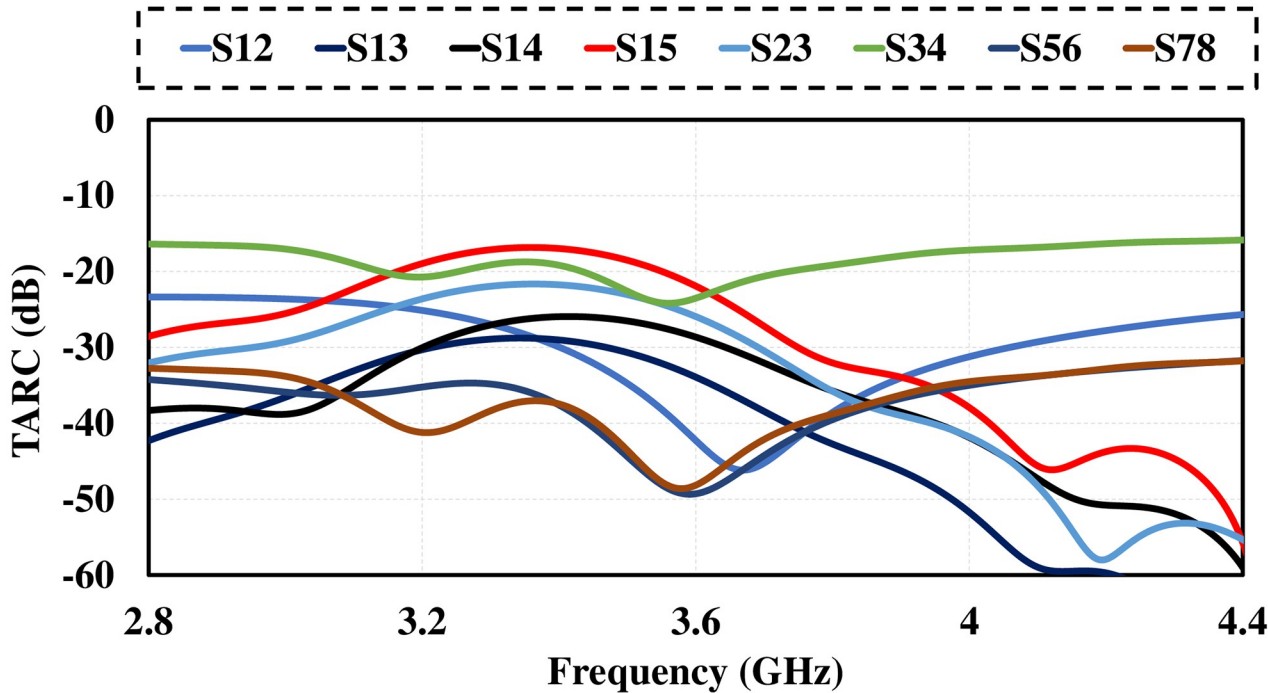

**Fig 17. TARC.** TARC of 8 by 8 MIMO antenna.

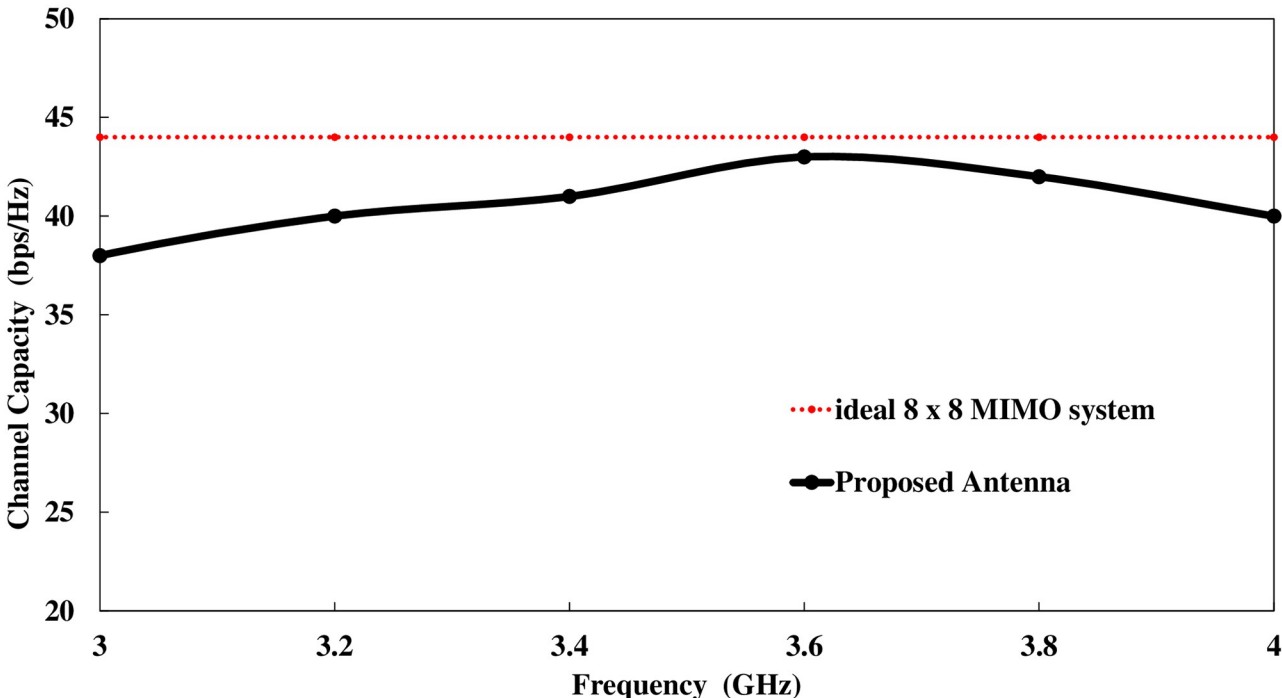

**Fig 18. Channel capacity.** Channel capacity of 8 by 8 MIMO antenna.

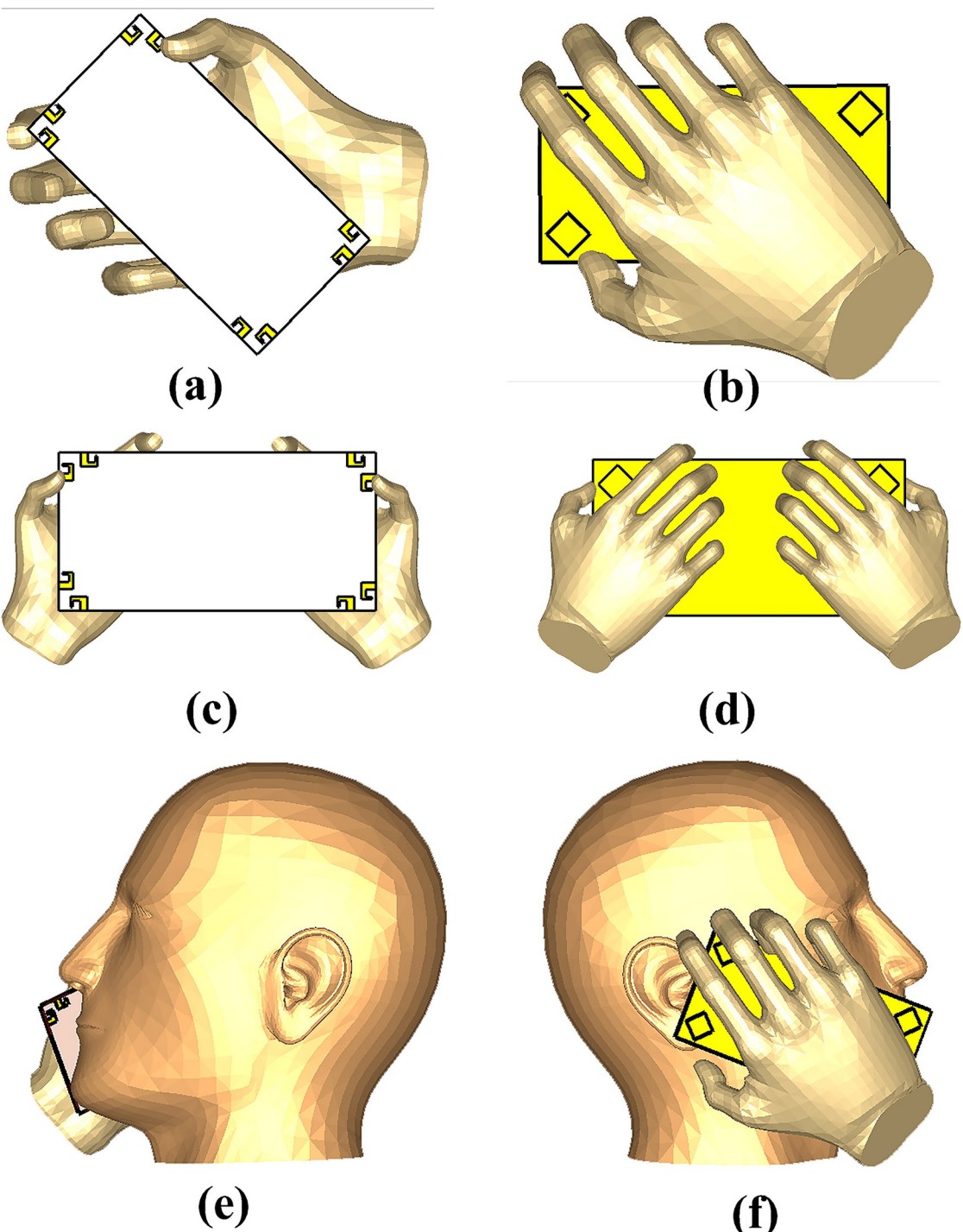

**Fig 19. Different posture.** A: SHM front View. B: SHM back View. C: DHM front View. D: DHM back View. E: TM front View. F: TM back View.

**Table 2. Comparison between proposed design and referenced 5G antennas.**

| Reference | Size (mm2) | Substrate Material | Design Method | Bandwidth (GHz) | Gain (dB) | Efficiency (%) | Isolation (dB) | Decoupling Method | ECC |
|---|---|---|---|---|---|---|---|---|---|
| [23] | 150 × 75 | Fr-4 | Circular-ring slot | 3.3-3.9 | >3 | 60-80 | 18 | Orthogonal Polarization | <0.005 |
| [24] | 150 × 70 | Fr-4 | Inverted L-shape | 2.5-3.6 | 2.3 | 45-65 | >10 | Slot in ground | <0.02 |
| [25] | 150 × 75 | Fr-4 | Double-sided parallel strip | 3.4-3.6 | 1.57 | 33-47 | 20 | Self-isolation | <0.4 |
| [26] | 150 × 80 | Fr-4 | T-shape | 2.54-2.62 | - | 52-70 | 10 | Polarization diversity | <0.1 |
| [27] | 136 × 68 | Fr-4 | Inverted L-shape | 3.3-3.7 | 4 | 50-75 | 15 | DGS | <0.1 |
| [28] | 150 × 75 | Fr-4 | Loop-type | 3.2-3.8 | 4 | 70-80 | 15 | Different operating frequency | <0.17 |
| [29] | 150 × 75 | Fr-4 | Square-ring slot | 3.4-3.8 | 3.9 | 60 | >20 | Orthogonal Polarization | <0.5 |
| [30] | 150 × 80 | Fr-4 | L-Shape | 3.4-3.6 | 2-4 | 60-80 | - | Different Frequency bands | <0.3 |
| [34] | 150 × 75 | Fr-4 | L-shaped | 3.3-3.9 | 3 | 60-80 | 17 | Orthogonal Polarization | <0.01 |
| [36] | 60 × 60 | Fr-4 | Jug-shaped CPW-fed | 3-11 | 3.4 | 68 | >20 | Orthogonal Polarization | <0.02 |
| Proposed | 150 × 75 | Fr-4 | Inverted J shaped diamond-slot | 3.3-4. 1 | 4.1 | 80-85 | >30 | DGS | <0.001 |

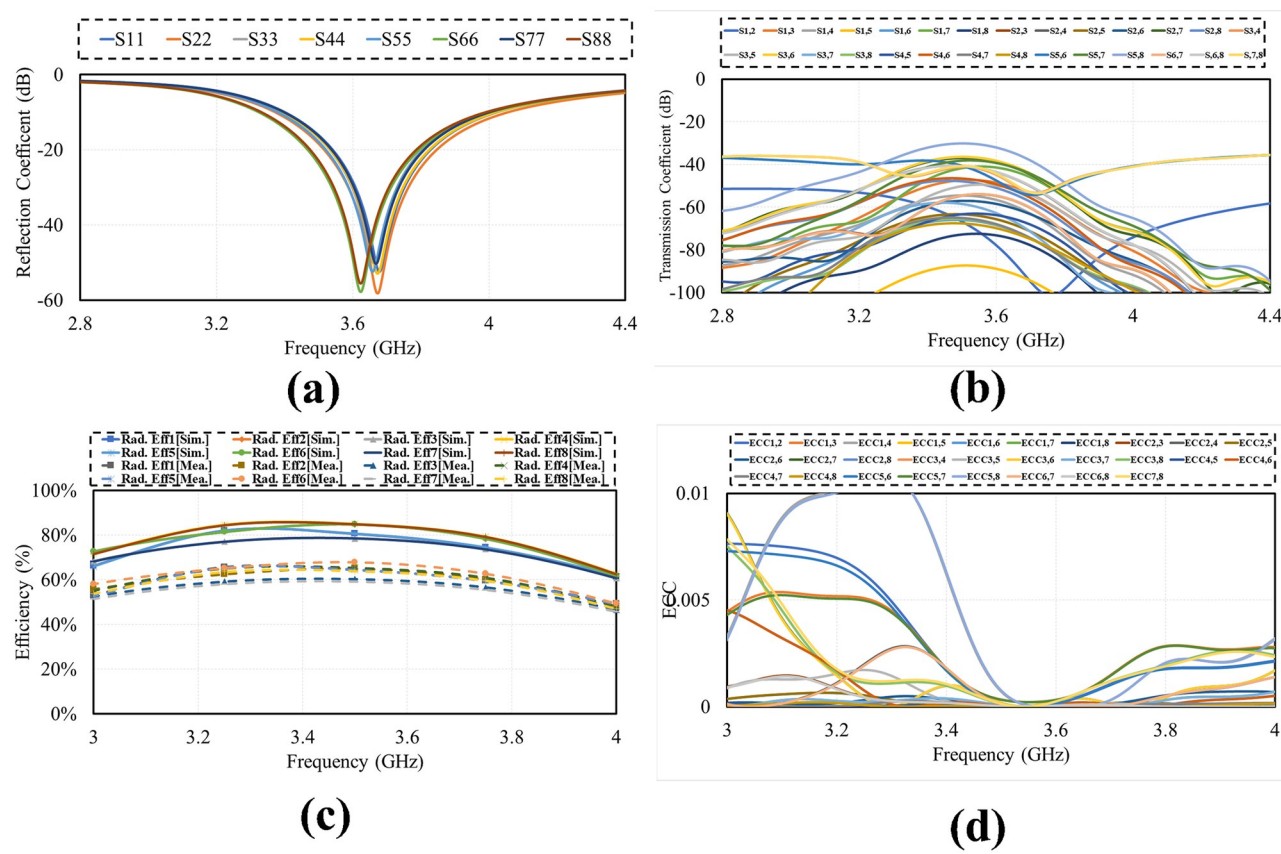

**Fig 20. SHM.** A: Reflection Coefficient. B: Tranmission Coefficient. C: Efficiency. D: ECC.

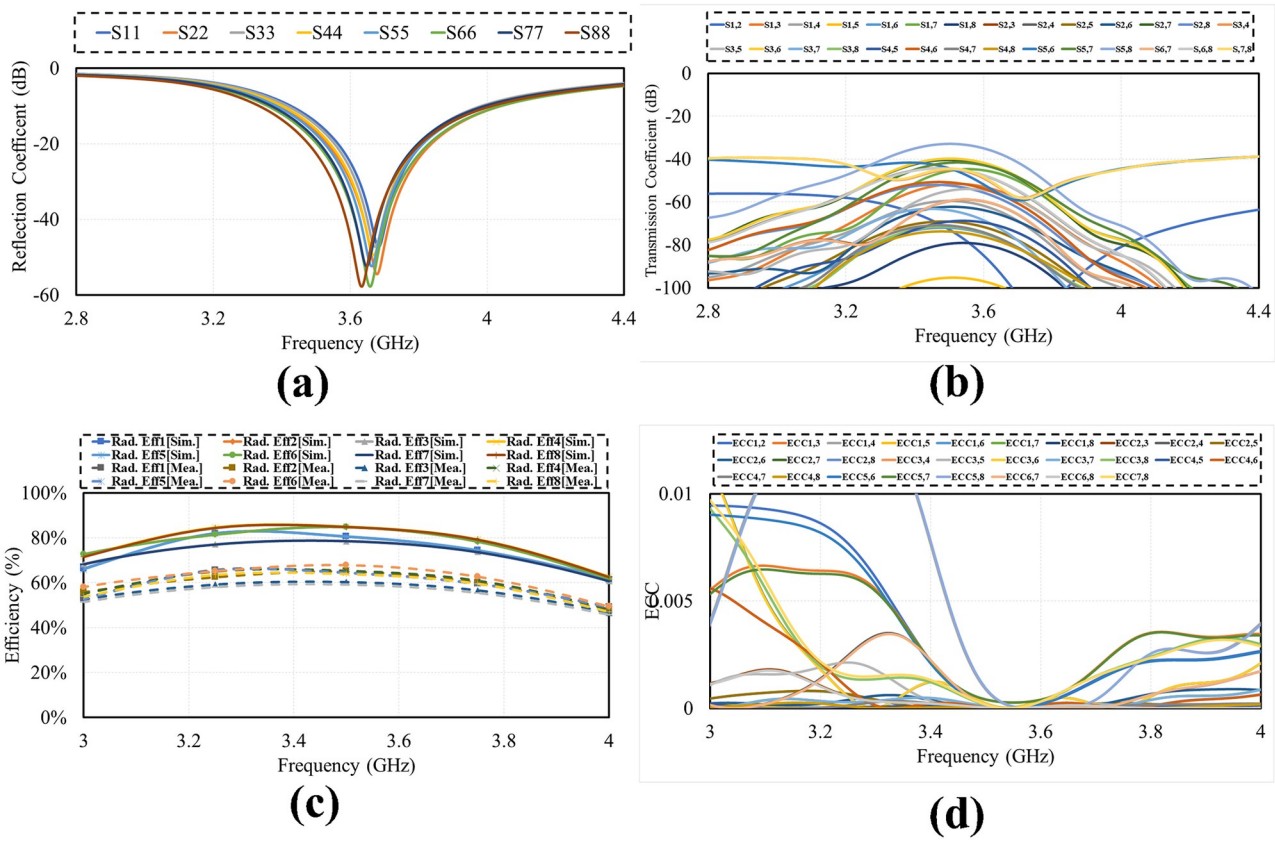

**Fig 21. DHM.** A: Reflection Coefficient. B: Tranmission Coefficient. C: Efficiency. D: ECC.

meant for mobile phone applications. The SAR measures the absorption level of electromagnetic waves in a human body. For our proposed design, the SAR characteristic of our proposed MIMO antenna system with user-head are investigated. The minimum and maximum SAR for the proposed MIMO antenna is 0.4 W/kg and 1.8 W/Kg respectively for Ant-5 and Ant-2. It can be concluded that the close distance between elements and the head phantom leads to a maximum SAR value and vice versa.

## Conclusion

For 5G MIMO communications, a mobile phone antenna with dual-polarization capabilities is suggested. The antenna layout includes two-port microstrip feed lines with ground plane diamond slots placed at each of the PCB's four corners. The antenna elements have a broad bandwidth with a 3.6 GHz center frequency. Antenna performance parameters i.e., S-parameters, Efficiency, maximum realized gain, radiation patterns, DG, ECC, TARC and channel capacity are simulated and adequate results are obtained. Antenna has more than 700 MHz bandwidth and radiation efficiency more than 85% with more than 6 dB gain for MIMO configuration. The antenna has exhibits more than 30dB isolation due to orthogonal placement of adjacent element. ECC one of main MIMO antenna parameters is less than 0.001. The antenna's performance in talk-mode and data-mode scenarios is examined. Also, the proposed design of smart-phone antenna is fabricated and tested. The experimental results are in decent agreement with simulated one's with very little variations due to experimental errors. The results

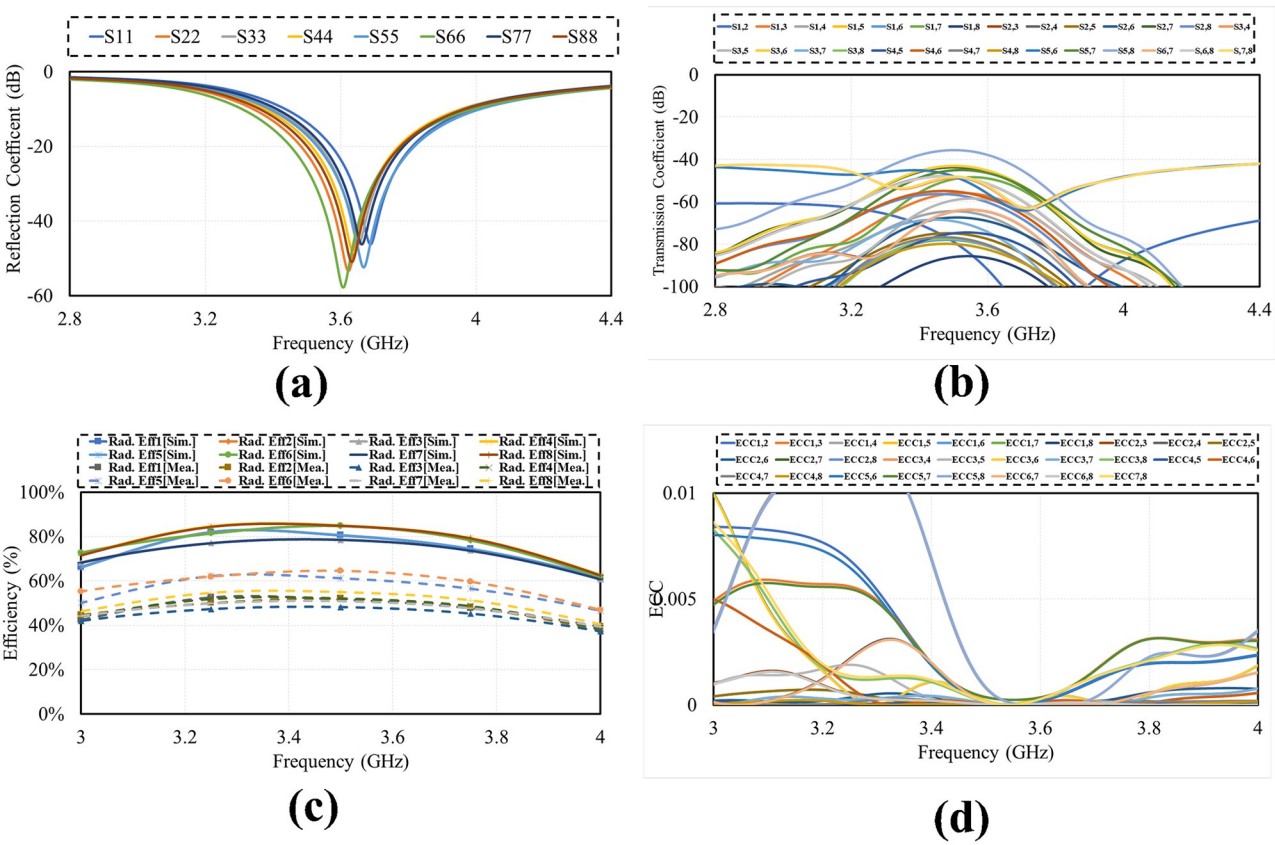

**Fig 22. Talk mode.** A: Reflection Coefficient. B: Tranmission Coefficient. C: Efficiency. D: ECC.

showed that the suggested smartphone antenna satisfies the criteria for use in upcoming smart-phones.

## Author Contributions

**Conceptualization:** Yasir Fawad, Abdulkarem H. M. Almawgani.

**Data curation:** Yasir Fawad, Muhammad Irfan, Rizwan Ullah.

**Formal analysis:** Yasir Fawad, Rizwan Ullah.

**Funding acquisition:** Muhammad Irfan, Saifur Rahman, Abdulkarem H. M. Almawgani, Salim Nasar Faraj Mursal.

**Investigation:** Muhammad Irfan, Rizwan Ullah, Fazal Muhammad, Abdulkarem H. M. Almawgani, Salim Nasar Faraj Mursal.

**Methodology:** Sadiq Ullah, Fazal Muhammad, Abdulkarem H. M. Almawgani, Salim Nasar Faraj Mursal.

**Project administration:** Sadiq Ullah, Saifur Rahman, Abdulkarem H. M. Almawgani.

**Resources:** Sadiq Ullah, Muhammad Irfan, Saifur Rahman.

**Software:** Yasir Fawad, Muhammad Irfan, Rizwan Ullah, Abdulkarem H. M. Almawgani, Salim Nasar Faraj Mursal.

**Supervision:** Sadiq Ullah, Saifur Rahman, Fazal Muhammad.

**Validation:** Yasir Fawad, Rizwan Ullah, Saifur Rahman, Salim Nasar Faraj Mursal.

**Visualization:** Muhammad Irfan.

**Writing – original draft:** Yasir Fawad.

**Writing – review & editing:** Sadiq Ullah, Muhammad Irfan, Saifur Rahman, Fazal Muhammad, Abdulkarem H. M. Almawgani, Salim Nasar Faraj Mursal.

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
