## [Decision Letter · Decision Letter 0]

28 Apr 2023

PONE-D-23-09968Dual Polarized 8-port Sub 6 GHz 5G MIMO Antenna for Smart Phone and Portable Wireless ApplicationsPLOS ONE

Dear Dr. Muhammad,

Thank you for submitting your manuscript to PLOS ONE. After careful consideration, we feel that it has merit but does not fully meet PLOS ONE’s publication criteria as it currently stands. Therefore, we invite you to submit a revised version of the manuscript that addresses the points raised during the review process. Please submit your revised manuscript by Jun 12 2023 11:59PM. If you will need more time than this to complete your revisions, please reply to this message or contact the journal office at plosone@plos.org. Please include the following items when submitting your revised manuscript:A rebuttal letter that responds to each point raised by the academic editor and reviewer(s). You should upload this letter as a separate file labeled 'Response to Reviewers'.A marked-up copy of your manuscript that highlights changes made to the original version. You should upload this as a separate file labeled 'Revised Manuscript with Track Changes'.An unmarked version of your revised paper without tracked changes. You should upload this as a separate file labeled 'Manuscript'.

We look forward to receiving your revised manuscript.

Kind regards,

Yuan-Fong Chou Chau

Academic Editor

PLOS ONE

Journal Requirements:

  "The authors are thankful to the Deanship of Scientific Research at Najran University for funding this work under the Research Groups (NU/ RG/SERC/11/9)."

Reviewers' comments:

Reviewer's Responses to Questions

**Comments to the Author**

1. Is the manuscript technically sound, and do the data support the conclusions?

Reviewer #1: Yes

Reviewer #2: Yes

Reviewer #3: Partly

2. Has the statistical analysis been performed appropriately and rigorously? 

Reviewer #1: Yes

Reviewer #2: N/A

Reviewer #3: N/A

3. Have the authors made all data underlying the findings in their manuscript fully available?

Reviewer #1: Yes

Reviewer #2: Yes

Reviewer #3: Yes

4. Is the manuscript presented in an intelligible fashion and written in standard English?

Reviewer #1: Yes

Reviewer #2: Yes

Reviewer #3: No

5. Review Comments to the Author

Reviewer #1: This research work demonstrated and practically implemented a mobile phone antenna with dual-polarization capabilities which is a potential candidate for 5G MIMO communications. The antenna layout includes two-port microstrip feed lines with ground plane diamond slots placed at each of the PCB’s four corners. From this reviewer’s point of view promising results have been achieved and well discussed in the well-organized manuscript. So, the results have been experimentally validated and highlighted by providing a fair comparison with state-of-the-art. Although the concept and idea of this work were found interesting and they seem attractive for the scientific society, authors are requested to carefully address the following comments to improve the quality of the manuscript prior to final recommendation.

1) Please add the applied design method of the proposed MIMO antenna to the title.

2) Please explain some information about the advantage of the proposed decoupling method in the abstract section.

3) Average radiation gain and efficiency can be added to the abstract section.

4) Introduction section can be improved by adding more explanations along with proper references. For example, more discussions on 5g MIMO antennas are requested. Also, to realize MIMO antennas, isolation between the radiation elements are very important which need to be discussed. There are various decoupling methods which can be briefly mentioned. Below are helpful suggestions.

(i) 5G MIMO antenna

"H-shaped Eight-Element Dual-band MIMO Antenna for Sub-6 GHz 5G Smartphone Applications", IEEE Access, vol. 10, pp. 85619-85629, 2022.

"An Innovative Antenna Array with High Inter Element Isolation for Sub-6 GHz 5G MIMO Communication Systems", Scientific Reports, 12, 7907, 2022.

"mmWave Four-Element MIMO Antenna for Future 5G Systems", Applied Sciences, 12(9), 4280, 2022.

"Uni-Planar MIMO Antenna for Sub-6 GHz 5G Mobile Phone Applications", Applied Sciences, 12(8), 3746, 2022.

"Multiple Elements MIMO Antenna System with Broadband Operation for 5th Generation Smart Phones", IEEE Access, vol. 10, pp. 38446-38457, 2022.

“Novel MIMO Antenna System for Ultra Wideband Applications”, Applied Sciences, 12(7), 3684, 2022.

“A high gain multiband offset MIMO antenna based on a planar log-periodic array for Ku/K-band applications”, Scientific Reports, 12, 4044 (2022).

"A Compact CPW-Fed Ultra-Wideband Multi-Input-Multi-Output (MIMO) Antenna for Wireless Communication Networks,", IEEE Access, vol. 10, pp. 25278-25289, 2022.

"Printed Closely Spaced Antennas Loaded by Linear Stubs in a MIMO Style for Portable Wireless Electronic Devices", Electronics, 10(22), 2848, 2021.

"MIMO Antenna System for Modern 5G Handheld Devices with Healthcare and High Rate Delivery" Sensors, 21(21), 7415, 2021.

(ii) Decoupling methods to realize MIMO antennas

"A Comprehensive Survey on "Various Decoupling Mechanisms with Focus on Metamaterial and Metasurface Principles Applicable to SAR and MIMO Antenna Systems"", IEEE Access, vol. 8, pp. 192965-193004, 2020.

“Study on Isolation and Radiation Behaviours of a 34×34 Array-Antennas Based on SIW and Metasurface Properties for Applications in Terahertz Band Over 125-300 GHz”, Optik, International Journal for Light and Electron Optics, Volume 206, March 2020, 163222.

"Isolation Enhancement of Densely Packed Array Antennas with Periodic MTM-Photonic Bandgap for SAR and MIMO Systems", IET Microwaves, Antennas & Propagation, Volume 14, Issue 3, February 2020, pp. 183 - 188.

"Surface Wave Reduction in Antenna Arrays Using Metasurface Inclusion for MIMO and SAR Systems", Radio Science, 54, 1067–1075, 2019.

"Mutual-Coupling Isolation Using Embedded Metamaterial EM Bandgap Decoupling Slab for Densely Packed Array Antennas", IEEE Access, vol. 7, pp. 5182–51840, April 29, 2019.

"Mutual Coupling Suppression Between Two Closely Placed Microstrip Patches Using EM-Bandgap Metamaterial Fractal Loading", IEEE Access, vol. 7, Page(s): 23606 – 23614, March 5, 2019.

"Interaction Between Closely Packed Array Antenna Elements Using Metasurface for Applications Such as MIMO Systems and Synthetic Aperture Radars", Radio Science, Volume53, Issue11, November 2018, Pages 1368-1381.

“Antenna Mutual Coupling Suppression Over Wideband Using Embedded Periphery Slot for Antenna Arrays”, Electronics, 2018, 7(9), 198.

“Study on Isolation Improvement Between Closely Packed Patch Antenna Arrays Based on Fractal Metamaterial Electromagnetic Bandgap Structures”, IET Microwaves, Antennas & Propagation, Volume 12, Issue 14, 28 November 2018, p. 2241 – 2247.

“Meta-surface Wall Suppression of Mutual Coupling between Microstrip Patch Antenna Arrays for THz-band Applications”, Progress in Electromagnetics Research Letters, Vol. 75, page 105-111, 2018.

5) Design process of the proposed single antenna should be elaborated in depth. Please explain why authors have realized diagonal rectangular slots on the back? How were its dimensions optimized?

6) Quality of the plots are poor, they need to be improved.

7) More discussions on the surface current distributions should be added.

8) The feeding mechanism of the MIMO antenna should be explained in detail.

9) In comparison table 2 please add the terms “applied design method, radiation gain, and design complexity” as well to make it more comprehensive.

10) Please extend the conclusion by adding more numerical results and achievements.

11) Reference part needs to be improved as per above mentioned suggestions.

Reviewer #2: 1. An 8-port MIMO antenna for smartphone application is proposed.

2. Author need top carry out a systematic literature review to establish the technical contribution and need for the proposed work.

Please refer below paper for the same: Design and Analysis of Wideband Flexible Self-Isolating MIMO Antennas for Sub-6 GHz 5G and WLAN Smartphone Terminals

3. Also authors should refer to some latest sub-6 GHz MIMO antennas and do a thorough comparison to establish the novel contribution:

a. Compact wideband four element optically transparent MIMO antenna for mm-wave 5G applications

b. Multiband hybrid MIMO DRA for Sub‐6 GHz 5G and WiFi‐6 applications

c. Dual-band and dual-polarization CPW Fed MIMO antenna for fifth-generation mobile communications technology at 28 and 38 GHz

d. Wideband flexible/transparent connected-ground MIMO antennas for sub-6 GHz 5G and WLAN applications

4. Improve the quality of all the figures (Use vector software like origin or MATLAB to plot the results for better quality)

5. Show the coordinate axis next to the antenna geometry.

6. In Fig 6(a), gain should be in dBi and not dB.

7. For such configuration, the number of input ports increases significantly.

8. Present selected data in some of the graphs for brevity of data.

9. Fig 15 is not at all clear.

10. Authors should carry out SAR analysis.

Reviewer #3: The article "Dual Polarized 8-port Sub 6 GHz 5G MIMO Antenna for Smart Phone and Portable

Wireless Applications" need major revision to process to next level

1) The design orientation and associated mathematical formulation is missing in the paper

2) Figure captions not given with numbers for review

3) For eight ports, the transmission coefficient results shown are of low quality and data in zig zag.

4) Human phantom model is used in the work for analysis. I have not found any type SAR readings and related matter.

5) The CCL and Diversity Gain and TARC parameters for MIMO was not discussed and presented

6) 2D plots with several combinations without any analysis is presented

6. PLOS authors have the option to publish the peer review history of their article (what does this mean?). If published, this will include your full peer review and any attached files.

Reviewer #1: No

Reviewer #2: No

Reviewer #3: **Yes: **Dr. B T P Madhav

---

## [Author Response · Author response to Decision Letter 0]

20 Jun 2023

Original Manuscript ID: PONE-D-23-09968

Original Article Title: Dual Polarized 8-port Sub 6 GHz 5G MIMO Antenna for Smart Phone and Portable Wireless Applications

To: PLOS ONE Editor

Re: We are very thankful to the PLOS ONE journal team and reviewers for such a comprehensive and profound review. We have revised our manuscript in light of their valuable queries and suggestions. We hope our revision has improved the paper quality to a level of reviewers’ satisfaction. The answers to their specific suggestions/queries/comments are given below in detail.

Dear Editor,

Thank you for allowing a resubmission of our manuscript, with an opportunity to address the reviewers’ comments.

We are uploading (a) our point-by-point response to the comments (below) (response to reviewers), (b) an updated manuscript with yellow highlighting indicating changes (Revised Manuscript with Track Changes), and (c) a clean updated manuscript without highlights (Manuscript) (PDF main document).

Best regards,

Dr. Fazal Muhammad

Corresponding Author

 

Response to Reviewer #1: We would like to thank the reviewer for careful and thorough reading of this manuscript and for the thoughtful comments and constructive suggestions, which help to improve the quality of this manuscript. Our response follows (the reviewer’s comments are in italics). General Comments: This research work demonstrated and practically implemented a mobile phone antenna with dual-polarization capabilities which is a potential candidate for 5G MIMO communications. The antenna layout includes two-port microstrip feed lines with ground plane diamond slots placed at each of the PCB’s four corners. From this reviewer’s point of view promising results have been achieved and well discussed in the well-organized manuscript. So, the results have been experimentally validated and highlighted by providing a fair comparison with state-of-the-art. Although the concept and idea of this work were found interesting and they seem attractive for the scientific society, authors are requested to carefully address the following comments to improve the quality of the manuscript prior to final recommendation.

Reviewer #1 Concerns:

1. Please add the applied design method of the proposed MIMO antenna to the title.

2. Please explain some information about the advantage of the proposed decoupling method in the abstract section. 

3. Average radiation gain and efficiency can be added to the abstract section

4. Introduction section can be improved by adding more explanations along with proper references. 

5. Design process of the proposed single antenna should be elaborated in depth. Please explain why authors have realized diagonal rectangular slots on the back? How were its dimensions optimized?

6. Quality of the plots are poor; they need to be improved.

7. More discussions on the surface current distributions should be added.

8. The feeding mechanism of the MIMO antenna should be explained in detail.

9. In comparison table 2 please add the terms “applied design method, radiation gain, and design complexity” as well to make it more comprehensive.

10. Please extend the conclusion by adding more numerical results and achievements.

11. Reference part needs to be improved as per above mentioned suggestions.

Additional Questions:

1. Is the manuscript technically sound, and do the data support the conclusions? Yes

2. Has the statistical analysis been performed appropriately and rigorously? Yes

3. Have the authors made all data underlying the findings in their manuscript fully available? Yes

4. Is the manuscript presented in an intelligible fashion and written in standard English? Yes

 

Reviewer#1, Concern#1: Please add the applied design method of the proposed MIMO antenna to the title.

Author response: Thanks for valuable suggestion, as per respected reviewer we have change the title of our manuscript and added design method to title. Our new title is as under

“Dual-polarized 8-port sub 6 GHz 5G MIMO diamond-ring slot antenna for smart phone and portable wireless applications”

Reviewer#1, Concern#2: Please explain some information about the advantage of the proposed decoupling method in the abstract section.

Author response: Thank you very much for the valuable suggestions, as per the honorable reviewer suggestions we have added the information about the advantage of the proposed coupling method in the abstract section, as highlight in the revised version of the manuscript.

Reviewer#1, Concern#3: Average radiation gain and efficiency can be added to the abstract section

Author response: 

Thanks for the suggestion , the average gain and efficiency are added to abstract section, and highlighted in marked version of manuscript.

Reviewer#1, Concern#4: Introduction section can be improved by adding more explanations along with proper references.

Author response: 

Thank you very much, as per the honorable suggestion, the introduction section is improved by adding more explanation with proper references, and the changes are highlighted in marked version.

Reviewer#1, Concern#5: Design process of the proposed single antenna should be elaborated in depth. Please explain why authors have realized diagonal rectangular slots on the back? How were its dimensions optimized?

Author response: Thanks for your comment, the design process is elaborated and some mathematical equations are included as shown in revised version. The diagonal slot in ground plane is used to reduce the mutual coupling between antenna elements and increase bandwidth. The dimensions are optimized by doing some parametric study.

Reviewer#1, Concern#6: Quality of the plots are poor; they need to be improved.

Author response: Thanks for very much for the comments , the quality of plots is improved in the revised version and highlighted the revised manuscript.

Reviewer#1, Concern#7: More discussions on the surface current distributions should be added.

Author response: Thank you very much, As per honorable reviewer directions, more discussion on surface currents is added as shown in highlighted version.

Reviewer#1, Concern#8: The feeding mechanism of the MIMO antenna should be explained in detail.

Author response: 

The feeding mechanism is explained in detail. As the honorable reviewer suggested. Thank you

Reviewer#1, Concern#9: In comparison table 2 please add the terms “applied design method, radiation gain, and design complexity” as well to make it more comprehensive.

Author response: 

Table 2 is updated as per respected reviewer comment, design method, gain, coupling techniques, and material used columns are added to the table. 

Reviewer#1, Concern#10: Please extend the conclusion by adding more numerical results and achievements.

Author response: 

Thank you very much, Conclusion is extended by adding more numerical results as per honorable reviewer.

Reviewer#1, Concern#11: Reference part needs to be improved as per above mentioned suggestions 

Author response: Reference part is improved according to honorable reviewer suggestion. As highlighted in the references section in the revised version.

 

Response to Reviewer #2: We would like to thank the reviewer for careful and thorough reading of this manuscript and for the thoughtful comments and constructive suggestions, which help to improve the quality of this manuscript. Our response follows (the reviewer’s comments are in italics). General Comments: An 8-port MIMO antenna for smartphone application is proposed.

Reviewer #2 Concerns:

1. Author needs to carry out a systematic literature review to establish the technical contribution and need for the proposed work. Please refer below paper for the same: Design and Analysis of Wideband Flexible Self-Isolating MIMO Antennas for Sub-6 GHz 5G and WLAN Smartphone Terminals

2. Also, authors should refer to some latest sub-6 GHz MIMO antennas and do a thorough comparison to establish the novel contribution.

3. Improve the quality of all the figures (Use vector software like origin or MATLAB to plot the results for better quality)

4. Show the coordinate axis next to the antenna geometry.

5. In Fig 6(a), gain should be in dBi and not dB.

6. For such configuration, the number of input ports increases significantly.

7. Present selected data in some of the graphs for brevity of data.

8. Fig 15 is not at all clear.

9. Authors should carry out SAR analysis.

Additional Questions:

1. Is the manuscript technically sound, and do the data support the conclusions? Yes

2. Has the statistical analysis been performed appropriately and rigorously? N/A

3. Have the authors made all data underlying the findings in their manuscript fully available? Yes

4. Is the manuscript presented in an intelligible fashion and written in standard English? Yes

 

Reviewer#2, Concern#1: Author needs to carry out a systematic literature review to establish the technical contribution and need for the proposed work.

Author response: Thanks for valuable suggestion, as per respected reviewer, the introduction section is improved and a systematic literature review is made with proper references. The changes are highlighted in revised marked version. 

Reviewer#2, Concern#2: Also, authors should refer to some latest sub-6 GHz MIMO antennas and do a thorough comparison to establish the novel contribution. 

Author response: Thanks for valuable suggestion, as per respected reviewer some of latest work is referred and compared with the proposed design. The changes are highlighted in revised marked version.

Reviewer#2, Concern#3: Improve the quality of all the figures (Use vector software like origin or MATLAB to plot the results for better quality)

Author response: 

Thank you very much, as per the honorable reviewer comments, the quality of figures is improved.

Reviewer#2, Concern#4: Show the coordinate axis next to the antenna geometry.

Author response: Thank you very much, as per the honorable reviewer, coordinate axis is added next to antenna geometry. See Figure 1 and 9.

Reviewer#2, Concern#5: In Fig 6(a), gain should be in dBi and not dB.

Author response: Thank you very much, as per respected reviewer valuable suggestion, the gain is plotted in dBi instead of dB. See Figure 6(a).

Reviewer#2, Concern#6: For such configuration, the number of input ports increases significantly.

Author response: Thank you very much for the quaries, yes the number of input ports is increased but for 5G the data rate is directly proportional to the number of antenna elements and number of ports. 

Reviewer#2, Concern#7: Present selected data in some of the graphs for brevity of data.

Author response: Thank you very much, as per the honorable reviewer valuable suggestion, some of the symmetrical data has been removed for clarity and brevity of data.

Reviewer#2, Concern#8: Fig 15 is not at all clear.

Author response: Thank you very much for your valuable comment, as per the honorable reviewer suggestion, we have replaced Fig no 15 with a clear one.

Reviewer#2, Concern#9: Authors should carry out SAR analysis.

Author response: Thank you very much for suggestion, as per the honorable review suggestion a detailed SAR analysis is done and the results are presented in revised version. 

 

Response to Reviewer #3: We would like to thank the reviewer for careful and thorough reading of this manuscript and for the thoughtful comments and constructive suggestions, which help to improve the quality of this manuscript. Our response follows (the reviewer’s comments are in italics). General Comments: The article "Dual Polarized 8-port Sub 6 GHz 5G MIMO Antenna for Smart Phone and Portable Wireless Applications" need major revision to process to next level

Reviewer #3 Concerns:

1. The design orientation and associated mathematical formulation is missing in the paper

2. Figure captions not given with numbers for review

3. For eight ports, the transmission coefficient results shown are of low quality and data in zig zag.

4. Human phantom model is used in the work for analysis. I have not found any type SAR readings and related matter.

5. The CCL and Diversity Gain and TARC parameters for MIMO was not discussed and presented

6. 2D plots with several combinations without any analysis is presented

Additional Questions:

1. Is the manuscript technically sound, and do the data support the conclusions? Partly

2. Has the statistical analysis been performed appropriately and rigorously? N/A

3. Have the authors made all data underlying the findings in their manuscript fully available? Yes

4. Is the manuscript presented in an intelligible fashion and written in standard English? No

 

Reviewer#3, Concern#1: The design orientation and associated mathematical formulation is missing in the paper.

Author response: Thank you very much for bringing our attention to this, as per the honorable reviewer, the mathematical equations are added to the paper.

Reviewer#3, Concern#2: Figure captions not given with numbers for review

Author response: Sorry for missing this, this time figure captions are added with numbers, Thank you for your comment.

Reviewer#3, Concern#3: For eight ports, the transmission coefficient results shown are of low quality and data in zig zag.

Author response: Thank you very much for the valuable suggestion, as per the honorable reviewer, the quality of plot is improved.

Reviewer#3, Concern#4: Human phantom model is used in the work for analysis. I have not found any type SAR readings and related matter.

Author response: Thank you very much for suggesting SAR analysis, SAR analysis is done and the results are presented in revised version.

Reviewer#3, Concern#5: The CCL and Diversity Gain and TARC parameters for MIMO was not discussed and presented

Author response: Thank you very much for suggesting to add some important MIMO parameters like TARC, DG and channel capacity. The results are calculated and are presented in revised version.

Reviewer#3, Concern#6: 2D plots with several combinations without any analysis is presented

Author response: Thank you very much for valuable suggestion analysis of 2D plots are added to revised version.

Kind Regards

Dr. Fazal Muhammad 

Corresponding Author

---

## [Decision Letter · Decision Letter 1]

4 Jul 2023

Dual Polarized 8-port Sub 6 GHz 5G MIMO Antenna for Smart Phone and Portable Wireless Applications

PONE-D-23-09968R1

Dear Dr. Muhammad,

We’re pleased to inform you that your manuscript has been judged scientifically suitable for publication and will be formally accepted for publication once it meets all outstanding technical requirements.

Kind regards,

Yuan-Fong Chou Chau

Academic Editor

PLOS ONE

Additional Editor Comments (optional):

Reviewers' comments:

Reviewer's Responses to Questions

**Comments to the Author**

1. If the authors have adequately addressed your comments raised in a previous round of review and you feel that this manuscript is now acceptable for publication, you may indicate that here to bypass the “Comments to the Author” section, enter your conflict of interest statement in the “Confidential to Editor” section, and submit your "Accept" recommendation.

Reviewer #1: All comments have been addressed

Reviewer #2: All comments have been addressed

2. Is the manuscript technically sound, and do the data support the conclusions?

Reviewer #1: Yes

Reviewer #2: Yes

3. Has the statistical analysis been performed appropriately and rigorously? 

Reviewer #1: Yes

Reviewer #2: N/A

4. Have the authors made all data underlying the findings in their manuscript fully available?

Reviewer #1: Yes

Reviewer #2: Yes

5. Is the manuscript presented in an intelligible fashion and written in standard English?

Reviewer #1: Yes

Reviewer #2: Yes

6. Review Comments to the Author

Reviewer #1: Appropriate modifications have been applied as per requested to improve the quality of the manuscript to an acceptable level.

Reviewer #2: 1. The comments give by the reviewers are addressed and implemented properly.

2. The manuscript is good to be published in its present form.

7. PLOS authors have the option to publish the peer review history of their article (what does this mean?). If published, this will include your full peer review and any attached files.

Reviewer #1: No

Reviewer #2: No

---

## [Editor Report · Acceptance letter]

18 Jul 2023

PONE-D-23-09968R1 

Dual-polarized 8-port sub 6 GHz 5G MIMO diamond-ring slot
antenna for smart phone and portable wireless applications 

Dear Dr. Muhammad:

I'm pleased to inform you that your manuscript has been deemed suitable for publication in PLOS ONE. Congratulations! Your manuscript is now with our production department. 

Kind regards, 

on behalf of

Dr. Yuan-Fong Chou Chau 

Academic Editor

PLOS ONE